# Microphysics regimes due to haze-cloud interactions: cloud oscillation and cloud collapse

Fan Yang[1], Hamed Fahandezh Sadi[2], Raymond A. Shaw[2], Fabian Hoffmann[3], Pei Hou[2], Aaron Wang[4], and Mikhail Ovchinnikov[4]

[1]Brookhaven National Laboratory, Upton, New York, USA
[2]Michigan Technological University, Houghton, Michigan, USA
[3]Ludwig-Maximilians-Universität München, Munich, Germany
[4]Pacific Northwest National Laboratory, Richland, Washington, USA

**Correspondence:** Fan Yang (fanyang@bnl.gov)

**Abstract.** It is known that aqueous haze particles can be activated into cloud droplets in a supersaturated environment. However, haze-cloud interactions have not been fully explored, partly because haze particles are not represented in most cloud-resolving models. Here, we conduct a series of large-eddy simulations of a cloud in a convection chamber using a haze-capable Eulerian-based bin microphysics scheme to explore haze-cloud interactions over a wide range of aerosol injection rates. Results show that the cloud is in a *slow microphysics* regime at low aerosol injection rates, where the cloud responds slowly to an environmental change and droplet deactivation is negligible. The cloud is in a *fast microphysics* regime at moderate aerosol injection rates, where the cloud responds quickly to an environmental change and haze-cloud interactions are important. More interestingly, two more microphysics regimes are observed at high aerosol injection rates due to haze-cloud interactions. *Cloud oscillation* is driven by the oscillation of the mean supersaturation around the critical supersaturation of aerosol due to haze-cloud interactions. *Cloud collapse* happens under weaker forcing of supersaturation where the chamber transfers cloud droplets to haze particles efficiently, leading to a significant decrease (collapse) of cloud droplet number concentration. One special case of cloud collapse is the haze-only regime. It occurs at extremely high aerosol injection rates, where droplet activation is inhibited, and the sedimentation of haze particles is balanced by the aerosol injection rate. Our results suggest that haze particles and their interactions with cloud droplets should be considered especially in polluted conditions.

## 1 Introduction

Atmospheric clouds play an important role in Earth's radiation balance and hydrological cycle. Their optical properties and precipitation efficiency are strongly influenced by cloud microphysical composition (e.g., droplet size and concentration) and processes (e.g., droplet formation and growth). It is known that cloud droplets in the atmosphere grow from aerosol particles, most of which contain water-soluble materials, such as sodium chloride or ammonium sulfate. These water-soluble aerosol particles first absorb water vapor in a subsaturated environment to become aqueous droplets (known as haze particles) through deliquescence. Haze particles can then be activated into cloud droplets in a sufficiently supersaturated environment (i.e. when relative humidity is higher than 100%). The supersaturation needed to activate cloud droplets depends on aerosol properties as

explained by Köhler theory (Twomey, 1959). Changes in aerosol properties from various anthropogenic and natural emissions can have a significant impact on clouds, thereby affecting the climate system substantially. So far, aerosol-cloud interaction remains one of the largest uncertainties in climate projection, partly because of the poor representation of cloud microphysical processes in models and incomplete understanding of those processes at the fundamental level (Morrison et al., 2020).

It is challenging to isolate the impact of aerosol on cloud properties and evolution in the real atmosphere, because cloud microphysics, dynamics, and thermodynamics are coupled in a complex way. In addition, cloud properties fluctuate over time and space, making them difficult to thoroughly sample and interpret. In contrast, the Michigan Tech convection cloud chamber, also known as the Pi chamber, can maintain a steady state cloud for several hours under well-controlled initial and boundary conditions (Chang et al., 2016). The Pi chamber produces a well-mixed supersaturated environment by maintaining a warm, humid bottom surface and a cool, humid top surface through Rayleigh-Bénard convection. The cloud is formed by continuously injecting aerosol particles into the supersaturated environment, and it can reach a steady state when the droplet activation rate is balanced by the droplet sedimentation rate. Cloud properties are controlled by aerosol properties (e.g., aerosol size, chemical composition, and injection rate) and boundary conditions (e.g., top and bottom temperatures – the driving factor to create a supersaturated environment). Steady-state cloud properties in the Pi chamber can be measured in great detail, which provides a unique opportunity to explore aerosol-cloud-turbulence interactions in well-controlled environments.

Previous Pi chamber experiments have shown that increasing aerosol injection rates result in higher cloud droplet number concentrations, smaller mean droplet radii, and narrower droplet size distributions (Chandrakar et al., 2016). These trends are consistent with results from cloud-resolving large-eddy simulations of the Pi chamber (Thomas et al., 2019). Krueger (2020) derived an analytical expression for the equilibrium cloud droplet size distribution in a turbulent cloud chamber with the assumption of uniform supersaturation. This analytic droplet size distribution, along with three others that account for supersaturation fluctuations in different ways, have been compared with measured droplet size distributions in the Pi chamber (Chandrakar et al., 2020). Results show that all four analytical droplet size distributions match the observed distribution reasonably well for monodisperse aerosol injection. However, none of them matched well for polydisperse aerosol injections. Chandrakar et al. (2020) argued that it might be due to the Ostwald ripening effect (Korolev, 1995; Jensen and Nugent, 2017; Yang et al., 2018), which is not considered in those analytical models. Recently, Shaw et al. (2023) developed a theoretical model to describe the microphysical state of well-mixed monodisperse droplets in cloudy Rayleigh-Bénard convection. The model predicts that $N_d \sim n_{in}$ and $q_l \sim n_{in}$ in the *slow microphysics* regime (i.e., at low aerosol injection rates), while $N_d \sim n_{in}^{5/3}$ and $q_l \sim n_{in}^{2/3}$ in the *fast microphysics* regime (i.e., at high aerosol injection rates), where $N_d$ is the droplet number concentration, $n_{in}$ aerosol injection rate, and $q_l$ liquid water mixing ratio. The slow microphysics regime refers to a relatively clean condition where the cloud would respond slowly to an environmental change, while the fast microphysics regime refers to a relatively polluted condition where the cloud would respond quickly to an environmental change. Pi chamber observations confirm the nonlinear relationship between $q_l$ and $n_{in}$ in the fast microphysics regime (see Fig. 7 in Shaw et al., 2023), but more investigations are needed to evaluate the theory and its ability to represent microphysical properties in a convection cloud chamber.

Besides cloud droplets, observations using a digital optical particle counter show the existence of haze particles with diameters down to 0.6 $\mu$m (detection limit) in the Pi chamber (Prabhakaran et al., 2020). Results from direct numerical simulations with Lagrangian aerosol/droplet microphysics show that haze particles undergo multiple activation and deactivation cycles in a convection chamber (MacMillan et al., 2022). However, previous theoretical studies do not include the haze activation process for simplification (Krueger, 2020; Chandrakar et al., 2020; Shaw et al., 2023). In addition, most previous Pi chamber simulations do not fully resolve haze particles, because in these simulations as well as in most atmospheric cloud simulations, droplets are formed directly from aerosol particles based on Twomey-type activation parameterizations (Twomey, 1959), in which aerosols are activated into cloud droplets if the environmental supersaturation is larger than aerosol's critical supersaturation (Thomas et al., 2019; Grabowski, 2020). Recently, Yang et al. (2023) developed a haze-capable bin microphysics scheme to simulate the Pi chamber by directly calculating the condensational growth of haze and cloud droplets, which naturally resolves droplet activation process without further parameterization. Simulations using this haze-capable bin scheme can capture haze droplet size distributions, aligning well with simulations from a Lagrangian microphysics scheme, with the latter serving as the "truth" because it does not suffer numerical diffusion during droplet growth and advection (Morrison et al., 2018; Grabowski et al., 2019). Results also show that the simulated cloud properties using the haze-capable bin microphysics scheme agree reasonably well with those using Twomey-type activation. We refer to the Twomey-type activation scheme as the CCN-based bin microphysics scheme, because it treats dry aerosols as Cloud Condensation Nuclei (CCN) which behave like cloud droplets immediately after the environmental supersaturation is larger than a critical supersaturation (i.e., without resolving the growth of haze particles). A good agreement between the haze-capable and CCN-based bin microphysics schemes suggests that if we are only interested in the cloud microphysical properties, we could still use Twomey-type activation parameterizations. However, only two aerosol injection rates were used in Yang et al. (2023), and thus, it is not clear whether results from the CCN-based bin microphysics scheme will always be similar to those from the haze-capable bin microphysics scheme, especially in a low supersaturation environment where haze-cloud interaction is important (e.g., Prabhakaran et al., 2020).

In this study, we conduct a series of large-eddy simulations of the Pi chamber using both CCN-based and haze-capable bin microphysics schemes over a wide range of aerosol injection rates. We aim to address the following questions:

(a) How do cloud microphysical properties change over a wide range of aerosol injection rates (for constant boundary conditions)?

(b) Do simulation results agree with previous theoretical studies?

(c) How important are haze-cloud interactions in the Pi chamber as well as in natural clouds?

Specifically, related to the question (a), we aim to explore how the steady-state supersaturation, mean droplet radius, $N_d$, and $q_l$ change with aerosol injection rate. For the question (b), we aim to evaluate steady-state droplet size distribution predicted in Krueger (2020) and Chandrakar et al. (2020), as well as slow and fast microphysics regimes predicted in Shaw et al. (2023). Related to question (c), we aim to know whether cloud properties simulated by the CCN-based bin microphysics scheme are always consistent with those from the haze-capable bin microphysics scheme, as indicated by Yang et al. (2023), or if haze-capable microphysics must be used for certain atmospheric conditions. Note that the Pi chamber could be connected to some simple cloud systems like fog or non-drizzling shallow-layer clouds. So what we learn about haze-cloud interactions can be

transferred. We want to understand the conditions under which haze-cloud interactions become important, connecting our work to a broader atmospheric science context.

## 2 Model description and setup

We employ SAM-Chamber to conduct large-eddy simulations of the Pi chamber in this study. SAM-Chamber is an adapted and modified version of the System for Atmospheric Modeling (SAM, Khairoutdinov and Randall, 2003) with the major changes in the consideration of four side walls and the top surface to represent the chamber boundary condition (detailed in Thomas et al., 2019). SAM-Chamber has been used to simulate the Pi chamber to explore several topics, including the impact of various bin microphysics and advection schemes on Pi chamber simulations (Yang et al., 2022), impact of supersaturation fluctuations on
droplet formation and growth (Prabhakaran et al., 2022; Anderson et al., 2023), development of a haze-capable microphysics scheme (Yang et al., 2023), investigation of drizzle initiation in larger convection chambers (Thomas et al., 2023; Wang et al., 2024c), glaciation of mixed-phase clouds (Wang et al., 2024a), and dual signatures of entrainment (Wang et al., 2024b). The SAM-Chamber employed in this study is the one used in Wang et al. (2024c), where the wall fluxes of momentum, sensible heat, and moisture are modeled in accordance with Monin-Obukhov Similarity Theory (MOST, Monin and Obukhov, 1954) as
before but with the following changes: (1) The roughness lengths for momentum ($z_0$), sensible heat ($z_t$), and moisture ($z_q$) are tuned to match the mean fluxes obtained in the direct numerical simulations. (2) The hydrostatic stability on the side walls is assumed to be neutral, as the buoyancy is parallel rather than normal to the side walls. More details on the wall modeling are addressed in Wang et al. (2024c, see Section 2 and Appendix B therein).

**Table 1.** Summary of model setup.

| Variable | Value |
| --- | --- |
| Bottom surface | $T_b = 300$ K, water-saturated |
| Top surface | $T_t = 280$ K, water-saturated |
| Sidewall | $T_w = 290$ K, water-saturated |
| Surface roughness | $z_0 = 0.75$ mm, $z_t = 0.619 z_0$, $z_q = 0.756 z_0$ (based on Wang et al., 2024c) |
| Resolution | 6.25 cm $\times$ 6.25 cm $\times$ 6.25 cm (32 $\times$ 32 $\times$ 16 grids) |
| Domain | 2 m $\times$ 2 m $\times$ 1 m (height) |
| Aerosol property | Sodium chloride (NaCl), $r_a = 62.5$ nm |
| Cloud microphysics scheme | CCN-based, Haze-capable (Yang et al., 2023) |
| Aerosol injection rate | $0.001 \sim 50$ cm$^{-3}$s$^{-1}$ (detailed in the text) |

The model setup is summarized in Table 1. The temperature of the bottom surface is set to be 300 K, the top surface to be
280 K, and the side walls to be 290 K. In previous SAM-Chamber simulations (Thomas et al., 2019; Yang et al., 2022, 2023), the side walls were set to be subsaturated such that the domain-averaged supersaturation without cloud is about 2.5% based on early chamber observations (Chandrakar et al., 2016). Subsaturated side walls serve as a sink for water vapor, tending to

evaporate droplets nearby. Sidewalls have been improved (i.e., closer to be water saturated) recently in the real Pi chamber, such that clouds can form at much smaller top and bottom temperature differences (Prabhakaran et al., 2020). In this study, all
surfaces are set to be saturated with respect to water. The impact of side wall conditions on cloud properties will be discussed later. The simulation domain is 2 m × 2 m × 1 m with 6.25 cm grid spacing in all three directions. This grid spacing falls in the inertial subrange, according to the direct numerical simulations with similar Reynolds number and Rayleigh number performed by Wang et al. (2024d).

To mimic continuous injection of salt particles, monodisperse sodium chloride aerosol particles with a dry radius of 62.5 nm
are added in each grid box after each time step, as in previous studies (Yang et al., 2022, 2023). Cloud droplet formation and growth by condensation are simulated using either a CCN-based or haze-capable bin microphysics scheme. Both schemes are two-moment bin microphysics schemes based on Chen and Lamb (1994), with some differences detailed in Yang et al. (2023) and summarized below. For the CCN-based bin microphysics scheme (referred to as the $CL_{CCN}$), droplet size distribution is represented by 33 mass-doubling bins starting from 1 $\mu$m radius. Dry aerosol particles stay in the aerosol category and they will
be moved to the first bin of the cloud category if the environmental supersaturation (in their grid box) is larger than the critical supersaturation of the aerosol (0.08% for a salt particle of 62.5 nm in radius). Solute and curvature effects are not considered for droplet growth by condensation. Note that such treatment of cloud microphysical processes – Twomey-type parameterization of droplet formation and neglect of solute and curvature effects on droplet growth – is quite common in atmospheric cloud simulations. For the haze-capable bin microphysics scheme (referred to as the $CL_{Haze}$), aqueous droplets (including haze and
cloud) are represented by 40 mass-doubling bins starting from 0.1 $\mu$m radius. Dry aerosol particles initially become haze with the equilibrium size at a relative humidity of 90% (same as in Yang et al., 2023). The growth of haze and cloud droplets via condensation is calculated explicitly with solute and curvature effects considered, and thus the activation process from haze particle to cloud droplet is naturally resolved. Following Yang et al. (2023), haze particles here refer to droplets with radii smaller than 1 $\mu$m which is the bin edge closest to the critical radius of the aerosol (0.92 $\mu$m). In this study, we consider the
solute and curvature effects for the growth of cloud droplets (radii larger than 1 $\mu$m) in both $CL_{CCN}$ and $CL_{Haze}$ schemes. The main difference between the $CL_{CCN}$ scheme and the $CL_{Haze}$ scheme is the way to handle droplet activation as detailed above. Although all chamber surfaces are saturated with respect to water, droplet deactivation by evaporation can still occur due to turbulent supersaturation fluctuations. For the $CL_{CCN}$ scheme, evaporated droplets will be moved to the aerosol category if their radii get smaller than 1 $\mu$m in radius (the deactivation process). For the $CL_{Haze}$ scheme, deactivated droplets remain as haze
particles. Efflorescence is not considered, and if haze particles are less than 0.1 $\mu$m in radius, they stay in the smallest droplet bin. In both schemes, droplets can only be lost through the bottom surface due to sedimentation, but not through the side walls.

Following the modeling studies by Yang et al. (2023) and Wang et al. (2024a, c, b), sodium chloride aerosol particles of a 62.5-nm radius are injected uniformly throughout the computational domain at a prescribed volumetric rate. A total of twenty-five aerosol injection rates ($n_{in}$) are employed to explore their impact on cloud properties. $n_{in}$ ranges from 0.001 to 50
cm$^{-3}$s$^{-1}$ in the following way: 0.001 to 0.005 cm$^{-3}$s$^{-1}$ every 0.001 cm$^{-3}$s$^{-1}$, 0.01 to 0.05 cm$^{-3}$s$^{-1}$ every 0.01 cm$^{-3}$s$^{-1}$, 0.1 to 0.5 cm$^{-3}$s$^{-1}$ every 0.1 cm$^{-3}$s$^{-1}$, 1.0 to 5.0 cm$^{-3}$s$^{-1}$ every 1.0 cm$^{-3}$s$^{-1}$, and 10.0 to 50.0 cm$^{-3}$s$^{-1}$ every 10.0 cm$^{-3}$s$^{-1}$. Note that 14 values of $n_{in}$ between 0.2 and 13 cm$^{-3}$s$^{-1}$ were used in recent Pi chamber experiments (see Fig. 7 in Shaw et al.,

2023), while only two values (0.25 and 2.5 cm$^{-3}$s$^{-1}$) were used in the Pi chamber simulations by Yang et al. (2023). Here, we cover a range of $n_{in}$ that can be achieved in the Pi chamber, while extending $n_{in}$ to represent extremely clean and polluted

conditions. Although these exceptionally small and large $n_{in}$ values might be difficult to achieve in the real chamber mainly due to the current limitations of aerosol injection, they are helpful to explore haze-cloud interactions in various microphysics regimes that will be discussed in the next section.

The time step is 0.02 s and the total simulation is one hour. The domain-averaged data are output every minute from the beginning of the simulation, while instantaneous 3-D data are output every five minutes in the second half of the simulation.

# 3 Results

## 3.1 Impact of aerosol injection rate on bulk cloud properties

Figure 1 shows the impact of $n_{in}$ on droplet mean radius ($r_d$), $N_d$, and $q_l$. Here, $q_l$ is the liquid water mixing ratio. Specifically, $r_d$ and $N_d$ are calculated only for cloud droplets whose radii are larger than 1 $\mu$m. $q_l = q_c + q_h$ for the CL$_{Haze}$ scheme where $q_c$ is cloud water mixing ratio (for droplets radii larger than 1 $\mu$m) and $q_h$ is haze water mixing ratio (for droplets radii smaller than

1 $\mu$m), while $q_l = q_c$ for the CL$_{CCN}$ scheme. Each dot in the figure represents a temporally averaged (over the second half an hour) and spatially averaged (over the whole domain) value for one aerosol injection rate when using either the CL$_{CCN}$ (black) or CL$_{Haze}$ (red) scheme. Results show that cloud microphysical properties based on these two schemes are similar, suggesting that using the Twomey-type activation parameterization is good enough to simulate bulk cloud properties, especially for $N_d$ and $q_l$.

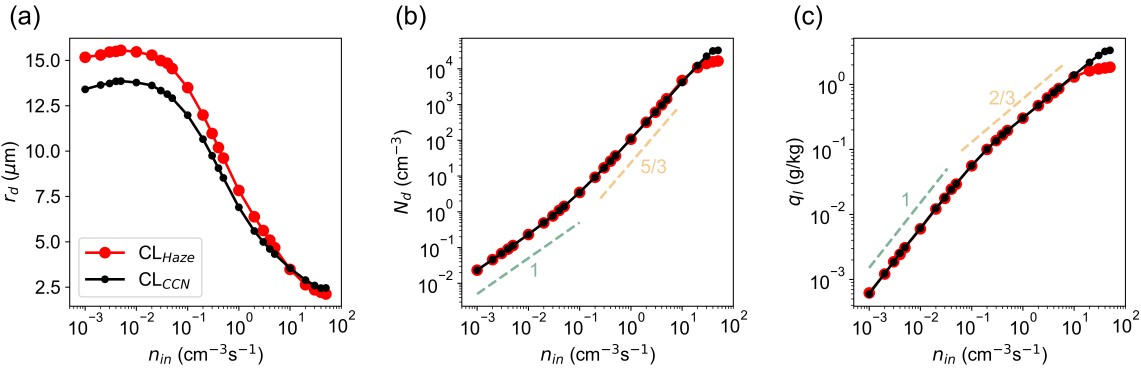

**Figure 1.** Spatial- (over the whole domain) and temporal-averaged (in the second half an hour) (a) mean droplet radius $r_d$, (b) droplet number concentration $N_d$, and (c) liquid water mixing ratio $q_l$ at various aerosol injection rates. Black and red dots are results using CL$_{CCN}$ and CL$_{Haze}$ schemes, respectively. Each dot represents the average of the variable over the whole domain from the second half an hour. The light green and yellow colored dashed lines in (b) and (c) are scaling relationships based on Shaw et al. (2023) in slow and fast microphysics regimes, respectively. Note that we only consider cloud droplets whose radii are larger than 1 $\mu$m to calculate $r_d$ and $N_d$ here.

The steady-state droplet size distributions based on the $CL_{CCN}$ and $CL_{Haze}$ schemes are shown in Fig. 2a-b. The distribution becomes narrower and shifts to smaller sizes with $n_{in}$, consistent with previous Pi chamber observations (Chandrakar et al., 2016) and simulations (Thomas et al., 2019; Yang et al., 2023). The mode of small haze particles can only be captured by the $CL_{Haze}$ scheme and is enhanced as $n_{in}$ increases (Fig. 2b). We also compare the simulated size distributions with four analytical droplet size distributions: $ar \exp(-br^4)$ (Fig. 2c), $ar \exp(-br^2)$ (Fig. 2d), $ar^2 \exp(-br^3)$ (Fig. 2e), and $a\sqrt{r} \exp(-br^3)$ (Fig. 2f), where $a$ and $b$ represent the combinations of other variables and parameters except for $r$. All these analytical distributions use steady-state $N_d$ and $q_c$ from the SAM-Chamber simulations as input to calculate the parameters $a$ and $b$. The precise formulas are displayed in Fig. 2 c-f. Chandrakar et al. (2020) detailed the assumptions regarding these analytical distributions and evaluated them with the Pi chamber observations. In short, $ar \exp(-br^4)$ is derived from the assumption of droplet growth in a constant supersaturation environment with size-dependent removal (Krueger, 2020), $ar \exp(-br^2)$ comes from droplet growth in a fluctuating supersaturation environment with size-independent removal (McGraw and Liu, 2006; Saito et al., 2019), $ar^2 \exp(-br^3)$ results from the principle of maximum entropy assumption (Liu and Hallett, 1998), and $a\sqrt{r} \exp(-br^3)$ comes from droplet growth in a fluctuating supersaturation environment with size-dependent removal (Chandrakar et al., 2020). Results show that the simulated cloud droplet size distributions are closer to $ar \exp(-br^4)$, $ar^2 \exp(-br^3)$, and $a\sqrt{r} \exp(-br^3)$, compared to $ar \exp(-br^2)$ which produces significantly broader spectra (Fig. 2d). Furthermore, the haze mode is not captured by any analytical distribution, simply because none of those analytical models considers the full activation process – from haze particles to cloud droplets.

Slow and fast microphysics regimes are observed as shown in Fig. 1. The impact of $n_{in}$ on the mean supersaturation $s$ and its standard deviation $\sigma_s$ (see Fig. 3) indicates the physical origin of these two microphysics regimes and its connection to various activation regimes. The slow microphysics regime is observed when $n_{in} < 0.1$ cm$^{-3}$s$^{-1}$. In this regime, few droplets (i.e., very small $N_d$ shown in Fig. 1b) grow in a high supersaturated environment (Fig. 3a) before they fall out, leading to a roughly constant $r_d$ (Fig. 1a) and a linear relationship between $n_{in}$ and $N_d$ (Fig. 1b) as well as $q_l$ (Fig. 1c) as predicted by Shaw et al. (2023). Based on the definition in Prabhakaran et al. (2020), the cloud is in the mean-supersaturation-dominated activation regime where $s >> s_{crit}$.

When $0.1$ cm$^{-3}$s$^{-1} < n_{in} < 10.0$ cm$^{-3}$s$^{-1}$, the cloud is in the fast microphysics regime, in which more cloud droplets compete with each other for available water vapor needed for their condensational growth, leading to larger $N_d$ and smaller $r_d$. In this regime, $r_d$, $s$, and $\sigma_s$ decrease with $n_{in}$, while $N_d \sim n_{in}^{5/3}$ and $q_l \sim n_{in}^{2/3}$, consistent with theory. Based on the definition in Prabhakaran et al. (2020), the cloud is in the supersaturation-fluctuation-influenced activation regime ($s > s_{crit}$ and $\sigma_s > s_{crit}$) or supersaturation-fluctuation-dominated activation regime ($s < s_{crit}$ and $\sigma_s > s_{crit}$), but the latter is barely observed in our results.

The scaling laws for $N_d$ and $q_l$ do not work well for $n_{in} \geq 10.0$ cm$^{-3}$s$^{-1}$ when using the $CL_{Haze}$ scheme (Fig. 1 b and c). Also note that both $s$ and $\sigma_s$ are smaller than $s_{crit}$ at these high aerosol injection rates, suggesting that droplet activation is strongly suppressed. It is interesting to see that $s$ approaches a value that is slightly smaller than $s_{crit}$ when using the $CL_{Haze}$ scheme, while in contrast, $s$ continuously decreases with $n_{in}$ and approaches 0 when using the $CL_{CCN}$ scheme. This is because the cloud system is buffered by a huge amount of cloud droplets in the polluted condition and $s$ should be close to the

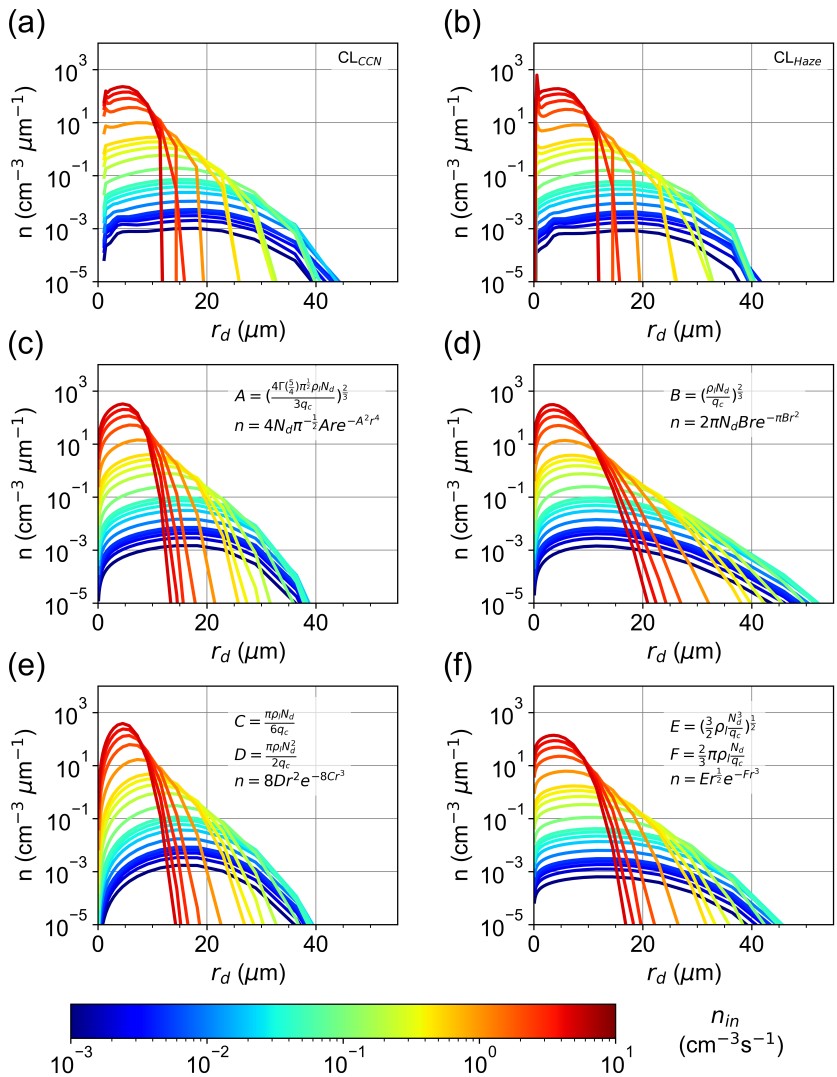

**Figure 2.** Steady-sate droplet size distributions for different aerosol injection rates when using (a) $CL_{CCN}$ and (b) $CL_{Haze}$ schemes. (c-f) Four analytical droplet size distributions using the domain-averaged $N_d$ and $q_c$ as input, with the precise formulas displayed in the legend.

equilibrium supersaturation over droplets (which is $s_{crit}$ when using the $CL_{Haze}$ scheme where solute and curvature effects are considered, or 0 when using the $CL_{CCN}$ scheme). This regime turns out to be very important for haze-cloud interactions which will be explored in the following section.

Table 2 summarizes the spatially and temporally averaged cloud microphysical properties for $n_{in} \leq 5.0$ cm$^{-3}$s$^{-1}$ when the scaling laws work reasonably well. Those variables include aerosol (when using the $CL_{CCN}$ scheme) / haze (when using the $CL_{Haze}$ scheme) number concentration ($N_a/N_h$), cloud droplet number concentration ($N_d$), mean cloud droplet radius

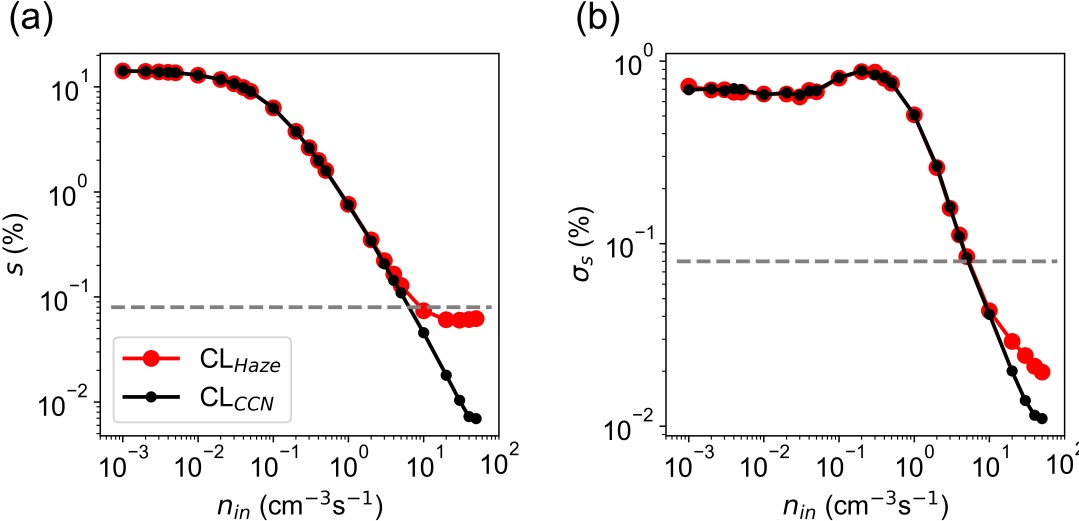

**Figure 3.** Spatial- (over the whole domain) and temporal-averaged (in the second half an hour) (a) mean supersaturation $s$ and (b) standard deviation of supersaturation $\sigma_s$ at various aerosol injection rates. Black and red dots are results using $CL_{CCN}$ and $CL_{Haze}$ schemes, respectively. Each dot represents the average of the variable over the whole domain from the second half an hour. The horizontal dashed line indicates the critical supersaturation of injected aerosols (0.08%).

($r_d$), droplet activation rate ($R_{act}$), and deactivation rate ($R_{deact}$). The droplet activation rate represents the number of newly formed cloud droplets per cubic centimeter per second, while the deactivation rate represents the reverse process. Note that the net activation rate ($R_{act} - R_{deact}$, the last two columns in Table 2) is close to $n_{in}$ (the first column in Table 2) for each case suggesting that the cloud reaches a quasi-steady state. It is worth mentioning that although the simulated cloud properties using

the two schemes are similar, unactivated particle concentration ($N_a$ or $N_h$), $R_{act}$, and $R_{deact}$ are quite different for $n_{in} \geq 1.0$ cm$^{-3}$s$^{-1}$. Our results suggest that haze-cloud interactions are important in the fast microphysics regime. The transition from the slow to the fast microphysics regime occurs when haze particles become important: $N_h/N_d > 5\%$ and $R_{deact}/R_{act} > 3\%$ for $n_{in} \geq 1.0$ cm$^{-3}$s$^{-1}$ (Fig. 4).

    Shaw et al. (2023) predicted that the transition from slow to fast microphysics regimes occurs at Da $\approx 1$. Here Da is the

Damköhler number, defined as the ratio of turbulent mixing time ($\tau_m$) to phase relaxation time ($\tau_p$) (see Eq. 1 in Lehmann et al., 2009). $\tau_p$ is inversely proportional to the product of $N_d$ and $r_d$, which can be determined from our simulation results. Take $n_{in} = 0.1$ cm$^{-3}$s$^{-1}$ as an example, $\tau_p \approx 70$ s, calculated from $N_d = 3.5$ cm$^{-3}$ and $r_d = 12$ $\mu$m based on Table 2 (using Eq. 18 in Korolev and Mazin, 2003). The apparent transition between slow and fast regimes as shown in Fig. 1 provides an opportunity to estimate $\tau_m$, which is about 70 s for our boundary conditions (e.g., 20 K difference in top and bottom temperature), if we

assume the transition occurs at Da $\approx 1$. However, this value is larger than another estimate of $\tau_m$ via $\tau_m = H/v_{air}$. Here, $H = 1$ m is the chamber height and $v_{air} \approx 0.1$ m s$^{-1}$ is the characteristic air speed in the chamber based on LES, leading to

**Table 2.** Spatial and temporal averaged aerosol/haze number concentration ($N_a$ or $N_h$, cm$^{-3}$), cloud droplet number concentration ($N_d$, cm$^{-3}$), mean cloud droplet radius ($r_d$, $\mu$m), droplet activation rate ($R_{act}$, cm$^{-3}$s$^{-1}$), and droplet deactivation rate ($R_{deact}$, cm$^{-3}$s$^{-1}$) at different aerosol injection rates ($n_{in}$, cm$^{-3}$s$^{-1}$). Values before and after the slash are results when using the CL$_{CCN}$ and CL$_{Haze}$ schemes, respectively. Each value is averaged over the whole domain in the second half an hour at a given $n_{in}$.

| $n_{in}$ (cm$^{-3}$s$^{-1}$) | $N_a$ or $N_h$ (cm$^{-3}$) | $N_d$ (cm$^{-3}$) | $r_d$ ($\mu$m) | $R_{act}$ (cm$^{-3}$s$^{-1}$) | $R_{deact}$ (cm$^{-3}$s$^{-1}$) |
|---|---|---|---|---|---|
| 0.001 | $0 / 1.0 \times 10^{-4}$ | 0.023 / 0.023 | 13 / 15 | 0.001 / 0.001 | 0 / 0 |
| 0.002 | $0 / 2.1 \times 10^{-4}$ | 0.046 / 0.046 | 14 / 15 | 0.002 / 0.002 | 0 / 0 |
| 0.003 | $0 / 3.2 \times 10^{-4}$ | 0.068 / 0.069 | 14 / 15 | 0.003 / 0.003 | 0 / 0 |
| 0.004 | $0 / 4.3 \times 10^{-4}$ | 0.092 / 0.091 | 14 / 15 | 0.004 / 0.004 | 0 / 0 |
| 0.005 | $0 / 5.5 \times 10^{-4}$ | 0.11 / 0.11 | 14 / 16 | 0.005 / 0.005 | 0 / 0 |
| 0.01 | 0 / 0.0012 | 0.23 / 0.23 | 14 / 15 | 0.01 / 0.01 | 0 / 0 |
| 0.02 | 0 / 0.0026 | 0.49 / 0.49 | 14 / 15 | 0.02 / 0.02 | 0 / 0 |
| 0.03 | 0 / 0.0043 | 0.77 / 0.77 | 13 / 15 | 0.03 / 0.03 | 0 / 0 |
| 0.04 | 0 / 0.0063 | 1.1 / 1.1 | 13 / 15 | 0.04 / 0.04 | 0 / 0 |
| 0.05 | 0 / 0.0086 | 1.4 / 1.4 | 13 / 15 | 0.05 / 0.05 | 0 / 0 |
| 0.1 | 0 / 0.026 | 3.5 / 3.5 | 12 / 13 | 0.1 / 0.1 | $0 / 8.8 \times 10^{-20}$ |
| 0.2 | $4.3 \times 10^{-7} / 0.095$ | 9.2 / 9.2 | 11 / 12 | 0.2 / 0.2 | $0 / 2.5 \times 10^{-7}$ |
| 0.3 | $7.4 \times 10^{-5} / 0.25$ | 17 / 17 | 9.7 / 11 | 0.3 / 0.3 | $8.4 \times 10^{-5} / 3.4 \times 10^{-5}$ |
| 0.4 | $8.8 \times 10^{-4} / 0.53$ | 26 / 26 | 9.1 / 10 | 0.4 / 0.4 | $0.0012 / 3.0 \times 10^{-4}$ |
| 0.5 | 0.0038 / 0.96 | 37 / 37 | 8.5 / 9.6 | 0.51 / 0.51 | 0.0048 / 0.0014 |
| 1 | 0.19 / 5.5 | 108 / 107 | 6.9 / 7.8 | 1.2 / 1 | 0.18 / 0.032 |
| 2 | 4.8 / 30 | 321 / 316 | 5.6 / 6.4 | 3.8 / 2.3 | 1.8 / 0.24 |
| 3 | 19 / 73 | 608 / 607 | 5 / 5.6 | 7 / 3.5 | 4.1 / 0.52 |
| 4 | 39 / 127 | 955 / 978 | 4.6 / 5.1 | 10 / 4.9 | 6.3 / 0.96 |
| 5 | 65 / 198 | $1.4 \times 10^3 / 1.4 \times 10^3$ | 4.3 / 4.7 | 13 / 7.1 | 8.2 / 2 |

$\tau_m$ on the order of 10 s. It is also larger than another estimate of $\tau_m = H^{2/3}/\epsilon^{1/3} \approx 6$ s, where $\epsilon$ is the energy dissipation rate (about 0.005 m$^2$s$^{-3}$ from the simulation).

## 3.2 Haze-cloud interactions in the polluted conditions

Figure 1 c and d show that $N_d$ and $q_l$ do not follow the aforementioned scaling laws for $n_{in} \geq 10$ cm$^{-3}$s$^{-1}$. In this section, we explore the reason for this departure and show that haze-cloud interaction in these extremely polluted conditions can lead to some new microphysics regimes, including cloud oscillation, cloud collapse, and haze only.

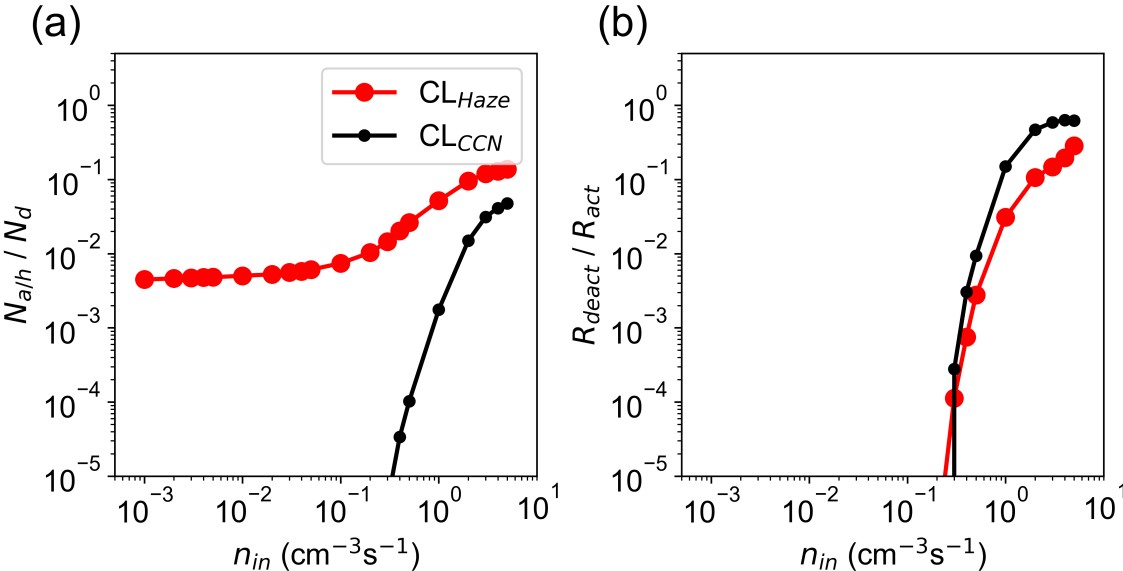

**Figure 4.** (a) The ratio of the unactivated particle number concentration to the cloud droplet number concentration for different aerosol injection rates ($n_{in}$). Unactivated particles are aerosol particles when using the $CL_{CCN}$ scheme, or haze particles when using the $CL_{Haze}$ scheme. (b) The ratio of deactivation to activation rate for different $n_{in}$.

### 3.2.1 Cloud oscillation

One possible reason for the observed departure for $N_d$ and $q_l$ in the polluted conditions ($n_{in} \geq 10$ cm$^{-3}$s$^{-1}$) is that the cloud does not reach a steady state after one hour. To rule out this possibility, we extend the simulations of the largest five $n_{in}$ (10, 20, 30, 40, 50 cm$^{-3}$s$^{-1}$) to a total simulation time of ten hours. Figure 5 shows the time series of domain-averaged $q_l$, $q_c$, $N_d$, $N_a$ (for the $CL_{CCN}$ scheme), $N_h$ (for the $CL_{Haze}$ scheme), total particle concentration ($N_T$), and $r_d$. Note that $q_l \geq q_c$ and $N_T = N_d + N_h$ when using the $CL_{Haze}$ scheme, and the difference ($q_l - q_c$) is haze water mixing ratio ($q_h$), while $q_l = q_c$ and $N_T = N_d + N_a$ when using the $CL_{CCN}$ scheme. Results show that $q_l$, $N_d$, and $r_d$ always reach a steady state when using the $CL_{CCN}$ scheme. Note that $N_a$ and $N_T$ increase with time for $n_{in} \geq 40$ cm$^{-3}$s$^{-1}$. This is because the sink of aerosol due to droplet activation is smaller than the source of aerosol due to aerosol injection, and thus aerosol particles accumulate. When using the $CL_{Haze}$ scheme, the cloud reaches a steady state for an aerosol injection rate of 10 cm$^{-3}$s$^{-1}$, where $q_l$ is dominated by $q_c$. In contrast, for $n_{in} \geq 20$ cm$^{-3}$s$^{-1}$, cloud microphysical properties (such as $q_l$, $q_c$, $N_d$, $r_d$) oscillate. The oscillation period increases as $n_{in}$ increases, and the periods are 15, 20, 25, and 30 min for $n_{in} = 20$, 30, 40, and 50 cm$^{-3}$s$^{-1}$. Meanwhile, the oscillation amplitude increases with $n_{in}$. $N_T$ has a much smaller oscillation magnitude compared with $N_d$ and $N_h$, suggesting that oscillations of $N_h$ and $N_c$ are out of phase. The local maximum of $N_d$ corresponds to the local minimum of $N_h$, indicating the burst of droplet formation is due to the activation of a large number of haze particles. The ratio of $q_h$ (i.e., $q_l$-$q_c$) to $q_l$ increases with $n_{in}$ and it can be up to 30% for $n_{in} = 50$ cm$^{-3}$s$^{-1}$. Note that the oscillation of the mean $r_d$ is mainly due to

droplet activation/deactivation, not due to the physical growth/evaporation of cloud droplets. For example, the rapid formation
of numerous small cloud droplets decreases the mean $r_d$ accordingly.

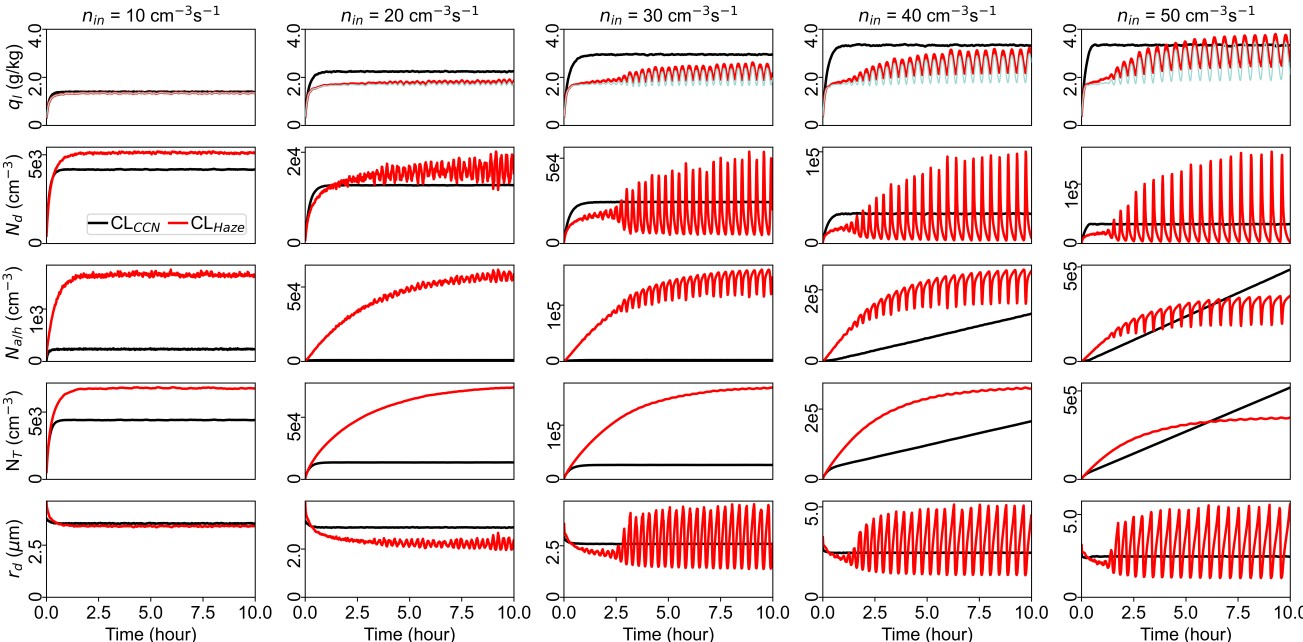

**Figure 5.** Time series of domain-averaged $q_l$ (first row), $N_d$ (second row), $N_a$ or $N_h$ (third row), $N_T$ (fourth row), and $r_d$ (fifth row) for five different $n_{in}$: 10, 20, 30, 40, and 50 cm$^{-3}$s$^{-1}$. The light blue line in the first row represents the cloud water mixing ratio ($q_c$) when using the CL$_{Haze}$ scheme.

Figure 6 shows time series of domain-averaged activation rate ($R_{act}$), deactivation rate ($R_{deact}$), supersaturation ($s$), standard deviation of supersaturation ($\sigma_s$), and surface precipitation rate ($P$). Here surface precipitation refers to the sedimentation of cloud droplets at the bottom surface. Results show that oscillations of bulk cloud properties when using the CL$_{Haze}$ scheme, as shown in Fig. 5, are associated with oscillations of process rates, like $R_{act}$, $R_{deact}$, and $P$. It is interesting to see that $s$ is close
to $s_{crit}$ (about 0.08%) when using CL$_{Haze}$ scheme, while $s$ decreases with $n_{in}$ and approaches 0 when using CL$_{CCN}$ scheme. This is because the cloud system is buffered by a huge amount of cloud droplets in the polluted condition and $s$ should be close to the equilibrium supersaturation over droplets. This equilibrium supersaturation is $s_{crit}$ when using the CL$_{Haze}$ scheme where solute and curvature effects are considered, but it is 0 when using the CL$_{CCN}$ scheme. Because $\sigma_s$ is much smaller than $s_{crit}$ at high injection rates, droplet activation is mainly controlled by the mean $s$. The oscillation of $s$ around the $s_{crit}$ leads to
the oscillation of droplet activation, and further causes the oscillation of cloud properties.

Figure 7 shows the time evolution of mean profiles of cloud properties in the last hour of the simulation for $n_{in} = 40$ cm$^{-3}$s$^{-1}$. We note that $q_c$ and $q_h$ oscillate out of phase (Fig. 7 a vs. d), while $q_l$ is mainly influenced by $q_c$ (Fig. 7 g). Larger $q_c$ ($q_h$) corresponds to smaller $N_d$ ($N_h$), and vice versa (Fig. 7 a vs. b and d vs. e). The anti-correlation between $q_c$ and $N_d$

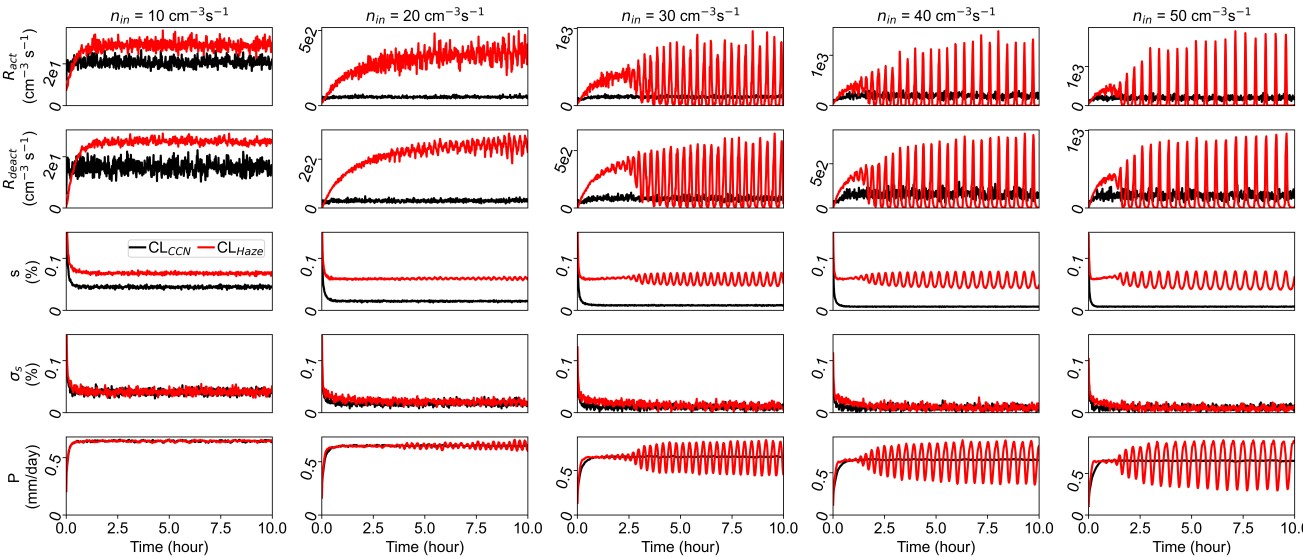

**Figure 6.** Same cases in Fig. 5, time series of domain-averaged $R_{act}$ (first row), $R_{deact}$ (second row), $s$ (third row), $\sigma_s$ (fourth row), and surface precipitation rate $P$ (fifth row) for five different $n_{in}$: 10, 20, 30, 40, and 50 cm$^{-3}$s$^{-1}$.

is opposite to their scaling relationships in the slow and fast microphysics, which are $q_c \sim N_d$ and $q_c \sim N_d^{2/5}$, respectively (Shaw et al., 2023). The sharp increase in $N_d$ (Fig. 7b) corresponds to a larger activation rate (Fig. 7c) due to the enhanced supersaturation (Fig. 7i), while the decrease in $N_d$ corresponds to a larger deactivation rate and a smaller supersaturation. To better show the low value of $R_{act}$ and $R_{deact}$, we constrain the range of $R_{act}$ and $R_{deact}$ to values below 240 cm$^{-3}$s$^{-1}$ if their values are larger than it when plotting Fig. 7 c and f. It can be seen that deactivation occurs in a much larger region (i.e., outside the top and bottom surfaces) and over a longer period within one cycle. However, the peak of $R_{act}$ is larger than the peak of $R_{deact}$ (see Fig. 6 first and second rows). The net activation rate ($R_{act} - R_{deact}$) averaged over one cycle should still be positive so that sedimentation is balanced by the net activation.

To further explore the mechanism of the oscillation, we pick one oscillation cycle for $n_{in} = 40$ cm$^{-3}$s$^{-1}$. Figure 8 shows the phase diagram of four pairs of variables: $N_h$ vs. $N_d$, $q_h$ vs. $q_c$, $q_h$ vs. $N_h$, and $q_c$ vs. $N_d$. Each circle in the figure represents the domain-averaged value at one time and its color represents the domain-averaged supersaturation with the unit of ‰₀, one per ten thousand. The size of the circle represents the mean droplet radius in a relative way: a larger circle means a larger $r_d$. The oscillation behavior can be explained by the circulation in the phase diagram. Taking Fig. 8d as an example: Start from the lower left corner where $q_c$ and $N_d$ are low, $s$ is high, and $r_d$ is large. When $s > s_{crit}$ ($s_{crit} \approx 8$ ‰₀ in this study), a huge amount of droplets are activated leading to a sharp increase in $N_d$. Newly formed cloud droplets significantly decrease the mean $r_d$ and they grow in slightly supersaturated conditions, leading to an increase in $q_l$ and a decrease in $s$. Shortly thereafter, $N_d$ decreases because droplet activation is suppressed when $s < s_{crit}$, and meanwhile, droplets are lost due to sedimentation and deactivation. Note that droplet loss is dominated by deactivation, and deactivation is driven by the mean supersaturation

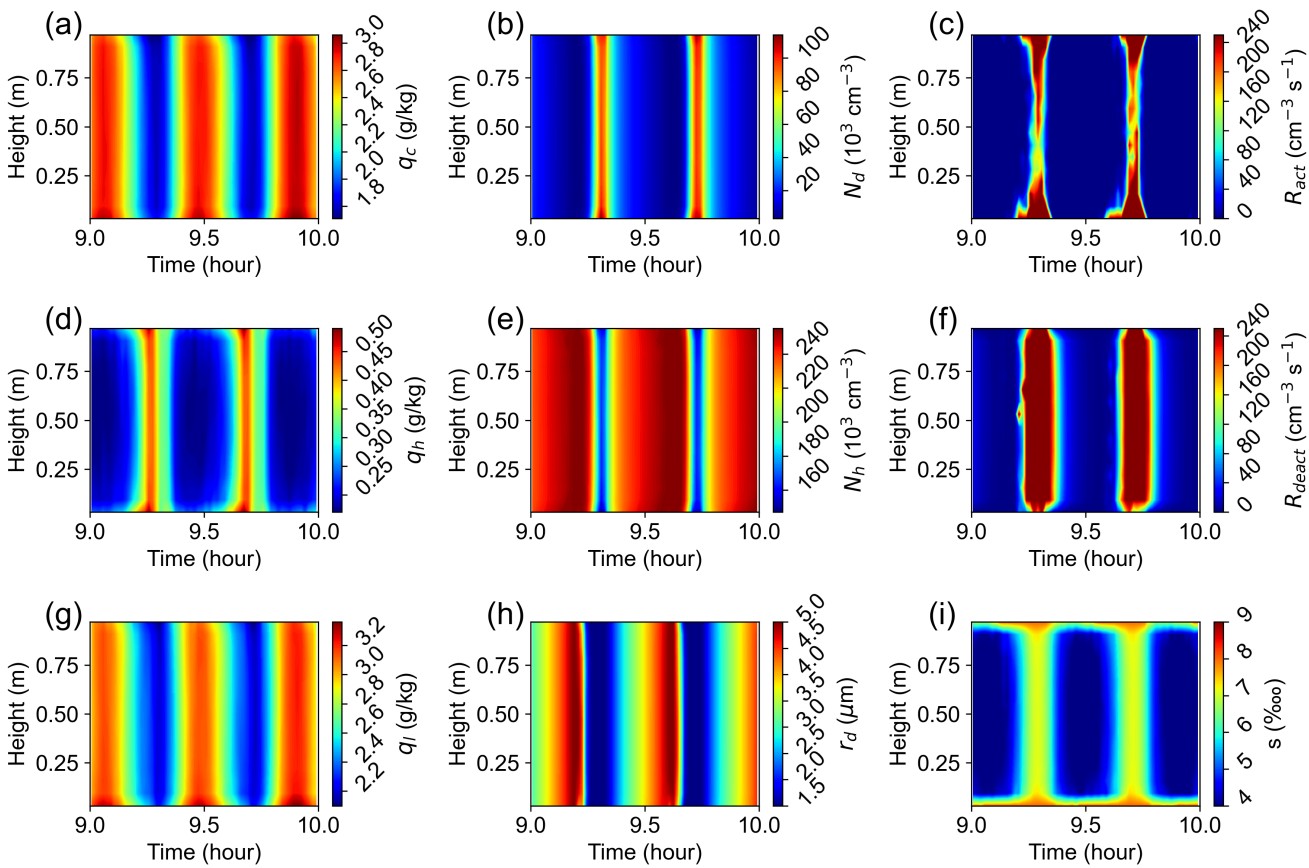

**Figure 7.** Time evolution of mean profiles of (a) cloud water mixing ratio, $q_c$, (b) cloud droplet number concentration, $N_d$, (c) activation rate, $R_{act}$, (d) haze water mixing ratio, $q_h$, (e) haze number concentration, $N_h$, (f) deactivation rate, $R_{deact}$, (g) total water mixing ratio, $q_l$, (h) droplet radius $r_d$, and (i) supersaturation, $s$, for $n_{in} = 40 \text{ cm}^{-3}\text{s}^{-1}$ between 9 and 10 hours when using the CL$_{\text{Haze}}$ scheme. It is the last simulation hour of Fig. 5, second column.

rather than supersaturation fluctuation because $s$ oscillates around $s_{crit}$ while $\sigma_s$ is much smaller than $s_{crit}$ as shown in Fig. 6. Droplet deactivation causes a recovery of $N_h$ and an increase in $q_h$ (Fig. 8 a,b). The decrease in $N_d$ finally results in a decrease in $q_l$ and an increase in $s$. When $s > s_{crit}$, another period starts. Note that droplet activation leads to an increase in $N_d$ and a decrease in $N_h$ simultaneously, thus causing the perfect anticorrelation between $N_h$ and $N_d$ (Fig. 8a). In contrast, mass and number concentrations (either $q_h$ vs. $N_h$ or $q_c$ vs. $N_d$) peak at different times, because it takes time for droplet/haze to grow. It is interesting to see that the oscillation evolves with time anticlockwise in $q_h - N_h$ diagram (Fig. 8c) and $q_c - N_d$ diagram (Fig. 8d), suggesting that the change in number concentration is ahead of the change in mass mixing ratio in their phases, analogous to a predator-prey dynamical system.

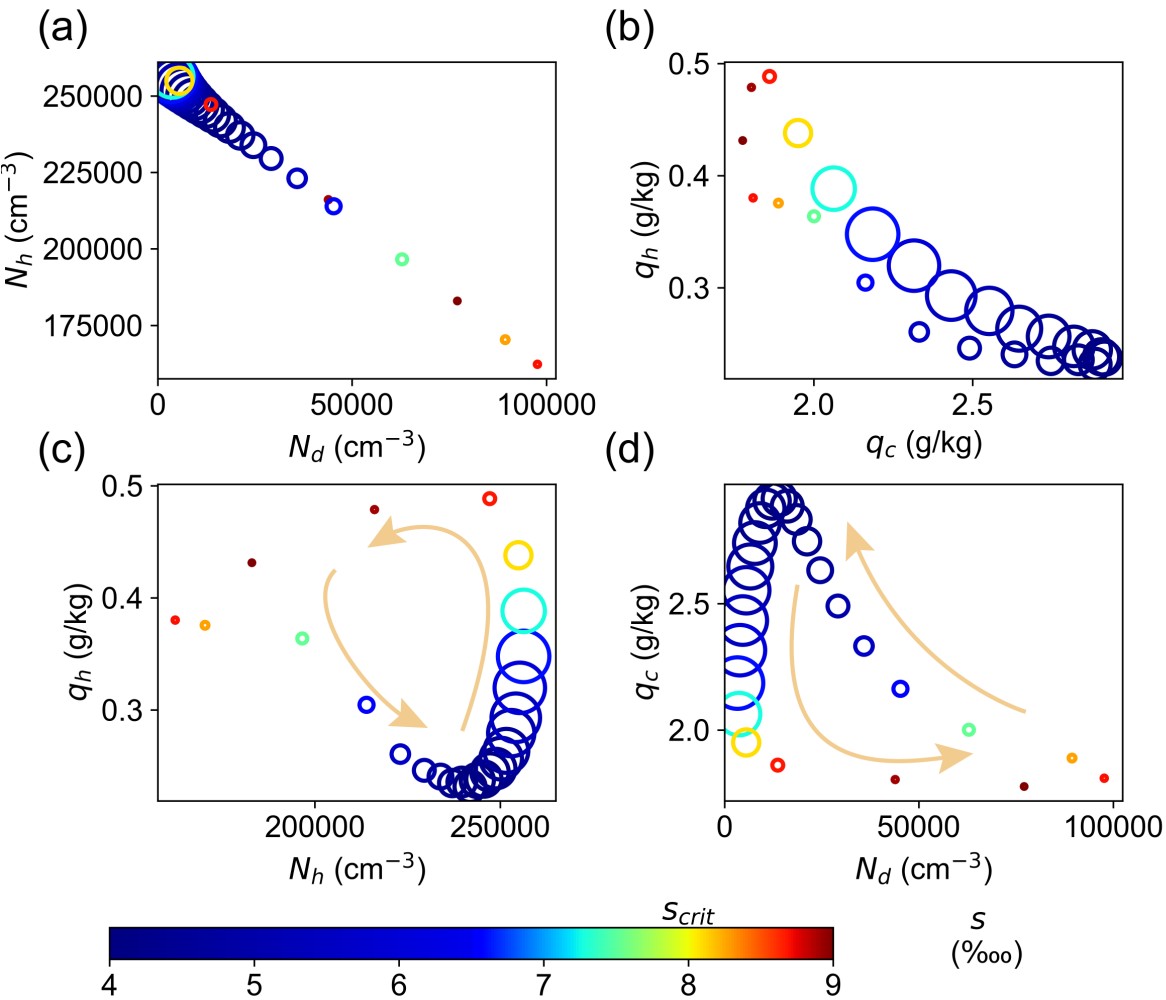

**Figure 8.** The relationship between domain-averaged (a) $N_h$ vs. $N_d$, (b) $q_h$ vs. $q_c$, (c) $q_h$ vs. $N_h$, and (d) $q_c$ vs. $N_d$ over one cycle of cloud oscillation at $n_{in} = 40\,\mathrm{cm}^{-3}\mathrm{s}^{-1}$. The size of the circle represents the mean droplet radius in a relative way, e.g., a larger circle means a larger $r_d$. Its color stands for the domain-averaged supersaturation with a unit of ‰₀₀, one per ten thousand. The arrows in (c) and (d) represent its time evolution in one cycle.

### 3.2.2 Cloud oscillation in a box model

To make sure the oscillation is physical and not due to numerical artifacts from using an Eulerian-based bin microphysics scheme, we develop a box model using a particle-based microphysics approach to simulate a cloud in a convection chamber. The particle-based treatment, analogous to the Lagrangian droplet method, directly calculates and tracks the time evolution of droplet size. The well-mixed cloud system can be described by a set of differential equations detailed below.

Following Shaw et al. (2023), the time derivative of mean air temperature can be expressed as

$$\frac{dT}{dt} = \frac{T_0 - T}{\tau_m} + \frac{L}{c_p} \frac{dq_l}{dt}\bigg|_{diff},$$ (1)

where $T_0$ is the reference temperature, which equals to the mean temperature in the chamber without cloud droplets. $L$ is the latent heat of vaporization of water and $c_p$ is the specific heat of air. $\tau_m$ is the mixing time scale, which quantifies how efficient $T$ can be restored to $T_0$. Similarly, the time derivative of the water vapor mixing ratio is expressed as

$$\frac{dq_v}{dt} = \frac{q_{v0} - q_v}{\tau_m} - \frac{dq_l}{dt}\bigg|_{diff},$$ (2)

where $q_{v0}$ is the reference water vapor mixing ratio, which equals the mean water vapor mixing ratio in a cloud-free condition assuming both top and bottom surfaces are saturated with respect to water. The last terms in Eqs. 1 and 2 represent the impact of vapor diffusional growth ($dq_l/dt|_{diff}$) of droplets on $T$ and $q_v$, respectively.

To be consistent with the model setup of large-eddy simulations, monodisperse dry aerosol particles with radii of 62.5 nm are added at a constant rate using a particle-based super droplet method. Specifically, one new super particle (hereafter referred to as particle) is added at a constant rate: every second for $n_{in} \leq 5$ cm$^{-3}$s$^{-1}$ or every 20 seconds for the largest five $n_{in}$ to save computational time. Each particle represents numerous real particles per unit volume. We refer to this as multiplicity, denoted hereafter as $n_i$, which represents the concentration of a particle with an index of $i$. Note that the multiplicity in this study is different from that in Lagrangian microphysics schemes (e.g., Shima et al., 2009; Hoffmann et al., 2015) in which it represents multiple number (instead of concentration) of identical droplets represented by the Lagrangian particle/superdroplet. The growth rate of droplet radius with an index of $i$ is given by,

$$\frac{dr_i}{dt} = \frac{G}{r_i}\left(s - \frac{A}{r_i} + \frac{B}{r_i^3}\right),$$ (3)

where $G$ is the growth factor and $s$ is the supersaturation depending on both $T$ and $q_v$. $A/r_i$ and $B/r_i^3$ are curvature and solute effects, respectively, in which $A$ and $B$ are constant for given thermodynamic and aerosol conditions (Eq. 6.6 in Yau and Rogers, 1996). The change in liquid water mixing ratio, which is linked to the last terms in Eqs. 1 and 2, can be calculated as the sum of mass change of all droplets,

$$\frac{dq_l}{dt}\bigg|_{diff} = \frac{4\pi\rho_l}{\rho_a} \sum_i n_i r_i^2 \frac{dr_i}{dt}.$$ (4)

Here $\rho_a$ and $\rho_l$ are air and liquid water densities, respectively.

Equations 1-4 are the governing equations to describe the bulk properties of a well-mixed cloud in a convection chamber. We use an ordinary differential equation solver to solve the above set of nonlinear and stiff equations (Brown et al., 1989). The total number of equations in the system depends on the number of particles. For example, if we have 100 particles at a given moment, the total number of equations to be solved is 102 (100 for $r_i$, one for $T$, and one for $q_v$). The same solver has been used in adiabatic cloud parcel models to properly calculate the growth of haze particles and the droplet activation process in the real atmosphere (Xue and Feingold, 2004; Chen et al., 2016; Yang et al., 2016).

Without sedimentation, the number of particles in the system would increase with time due to continuous injection, which eventually makes the system numerically unsolvable. In reality, the number of particles increases with time at the beginning, but it could reach a steady state if the rate of increase of particles due to injection is balanced by its loss rate due to sedimentation. To represent the impact of gravitational sedimentation, $n_i$ decreases with time as

$$\delta n_i = n_i \left( 1 - \exp \left[ -\frac{\delta t}{\tau_{sed}(r_i)} \right] \right) \approx n_i \frac{\delta t}{\tau_{sed}(r_i)}, \tag{5}$$

where $\delta t$ is set to be one second and $\delta n_i$ is the decreased amount of multiplicity of a particle with the index of $i$. $\tau_{sed}$ is the characteristic sedimentation time of a droplet with a radius of $r_i$ in a convection cloud chamber,

$$\tau_{sed}(r_i) = \frac{H}{v_t(r_i)}. \tag{6}$$

Here $H$ is the chamber height of 1 m and $v_t$ is the terminal velocity of a droplet with a radius of $r_i$. If $n_i$ is smaller than a threshold of $10^{-10}$ cm$^{-3}$, we remove that particle.

We conduct a total of 25 cases with the same forcing (i.e., $T_0$, $q_{v0}$, and $\tau_m$) but different $n_{in}$, which are the same as those used in previous large-eddy simulations. For a given $n_{in}$, the multiplicity of a newly added particle ($n_{i0}$) and the injection frequency are determined such that their product equals $n_{in}$. For example, injection of a particle with $n_{i0} = 0.5$ cm$^{-3}$ every second corresponds to $n_{in} = 0.5$ cm$^{-3}$s$^{-1}$, while injection of a particle with $n_{i0} = 200$ cm$^{-3}$ every 20 seconds corresponds to $n_{in} = 10$ cm$^{-3}$s$^{-1}$. $T_0$ and $q_{v0}$ are set to be 290 K and 13.9 g kg$^{-1}$, corresponding to a supersaturation of 15% in the

absence of all hydrometeors. This setup is consistent with the cloud-free humid condition in a convection chamber with a top temperature of 280 K, and a bottom temperature of 300 K, with both surfaces saturated with respect to water. $\tau_m$ is set to be 165 s, such that the steady-state $s$ from the box model (Fig. 9a) agrees with that from LES (Fig. 1a). Note that the value of $\tau_m$ used here is not the same as the estimated $\tau_m$ ($\sim 70$ s) for $Da = 1$ based on LES results above, but they are the same order of magnitude.

Results show that the impact of $n_{in}$ on cloud properties based on the box model is consistent with those from LES (compare Fig. 9 vs. Fig. 1). Slow and fast microphysics regimes are also captured by the box model (Fig. 9c,d). It is encouraging to see that the transition between slow and fast microphysics regimes occurs at around $n_{in}$ of 0.1 cm$^{-3}$s$^{-1}$, which agrees well with LES. The box model also captures cloud oscillation for the largest five $n_{in}$ (10, 20, 30, 40, 50 cm$^{-3}$s$^{-1}$), as shown in Figs. 10 and 11. The oscillation frequency decreases with the increase of $n_{in}$, which is consistent with LES results (Fig. 5). Note that,

for cloud oscillation cases, $s$, $r_d$, $N_d$, and $q_l$ in Fig. 9 are averaged over one cycle. It is interesting to see that $N_d$ vs. $n_{in}$ and $q_l$ vs. $n_{in}$ agree better with the aforementioned scaling laws in the fast microphysics regime, compared with LES (compare Fig. 9 c,d vs. Fig. 1 c,d). This might be due to the bias in representing droplet distribution when using a limited number of discretized bins in polluted conditions, or the systematic difference between a 3-D LES and a box model.

### 3.2.3   Origin of cloud oscillation

Results from LES and box models show the existence of cloud oscillation at high $n_{in}$, indicating that cloud oscillation is physically plausible, not due to numerical artifact. In this subsection, we discuss the physical origin of cloud oscillation and explain why the CL$_{\text{Haze}}$ scheme can simulate cloud oscillation, while the CL$_{\text{CCN}}$ scheme fails.

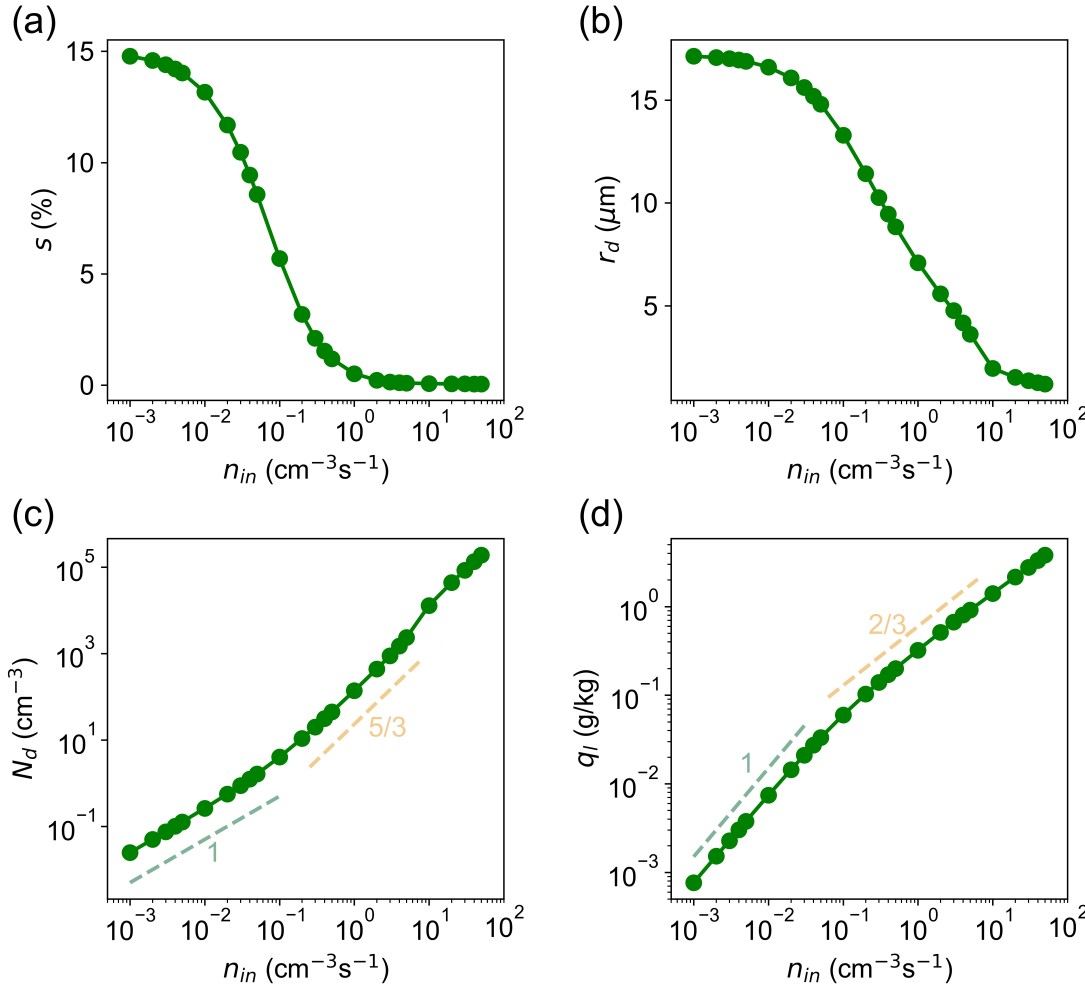

**Figure 9.** Impact of $n_{in}$ on (a) supersaturation $s$, (b) mean droplet radius $r_d$, (c) droplet number concentration $N_d$, and (d) liquid water mixing ratio $q_l$ based on a box model using a particle-based microphysics approach. Cloud oscillation occurs for the five largest $n_{in}$ values (10, 20, 30, 40, 50 cm$^{-3}$s$^{-1}$), as shown in Fig. 10. For those cases, $s$, $r_d$, $N_d$, and $q_l$ are averaged over one cycle. The light green and yellow colored dashed lines in (c) and (d) are scaling relationships based on Shaw et al. (2023) for slow and fast regimes, respectively.

Time series of $s$ shown in Figs. 6 and 11 provide more physical insights of cloud oscillation. The direct reason for cloud oscillation is that $s$ oscillates around $s_{crit}$ when using the CL$_{\text{Haze}}$ scheme. To be clear, cloud oscillation mentioned in this study

represents the oscillation of cloud bulk statistical properties. It is the oscillation of the whole well-mixed cloud system, not an individual droplet. The physical origin of cloud oscillation is due to the non-linear interactions between haze and cloud droplets in a dynamic system: (1) First, the supersaturation $s$ in the system is very close to $s_{crit}$, and most of the time $s < s_{crit}$. This

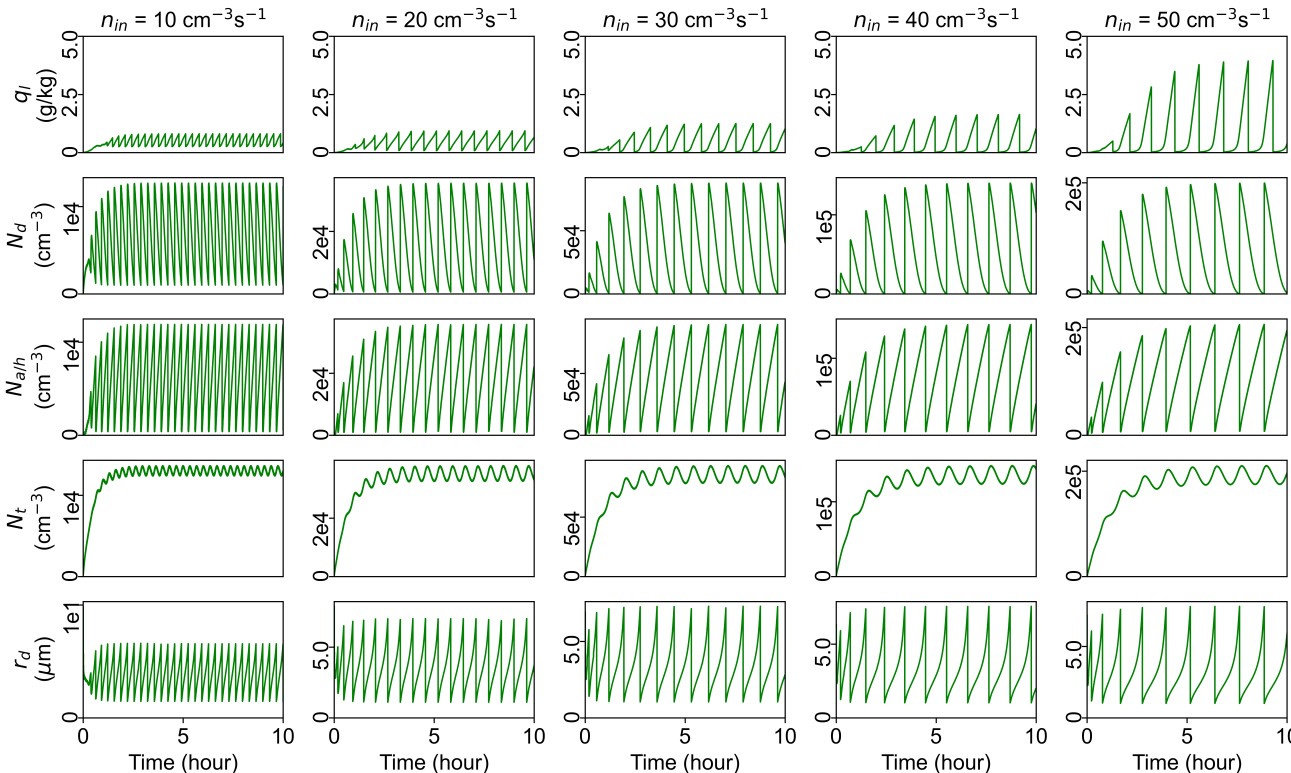

**Figure 10.** Time series of $q_l$ (first row), $N_d$ (second row), $N_h$ (third row), $N_t$ (fourth row), and $r_d$ (fifth row) from a box model using a Lagrangian microphysics approach for the five largest $n_{in}$: 10, 20, 30, 40, 50 cm$^{-3}$s$^{-1}$.

can happen in a heavily polluted condition where there are many haze particles. (2) There is a forcing in the system to maintain the supersaturation. In the Pi chamber, the forcing is due to the temperature difference between the top and bottom surfaces. In the real atmosphere, the forcing can be due to adiabatic cooling (e.g., in a rising cloud parcel) or radiative cooling (e.g., radiation fog). (3) When $s > s_{crit}$, a huge number of haze particles activate to cloud droplets and the consumption of water vapor due to droplet condensational growth leads to $s < s_{crit}$; (4) Under $s < s_{crit}$ condition, droplet activation is suppressed and droplet concentration decreases due to droplet deactivation and sedimentation; (5) Meanwhile, haze number concentration increases due to continuously aerosol injection and droplet deactivation. (6) $s$ increases with the decrease of the sink of water vapor due to fewer cloud droplets and more haze particles, and when $s > s_{crit}$, another cycle starts. In contrast, $s$ approaches 0 when using the CL$_{CCN}$ scheme (black line in the third row of Fig.6), suggesting that droplet activation is strongly suppressed in the bulk region.

Additionally, $\sigma_s$ decreases with $n_{in}$ and approaches 0 due to the buffering effect of cloud droplets under polluted conditions (Fig.6 fourth row). This suggests that droplet activation is controlled by the mean supersaturation instead of supersaturation fluctuation. This is why even though turbulence is not considered, the box model (Fig. 9) and the theoretical model (developed

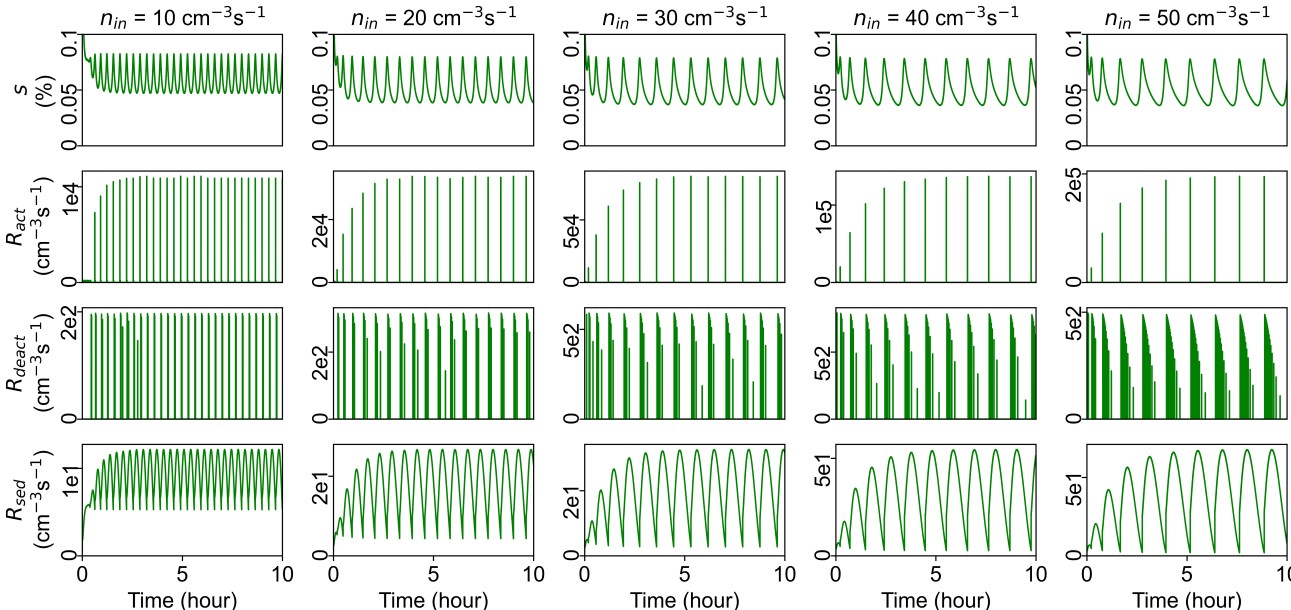

**Figure 11.** Similar to Fig. 10, time series of $s$ (first row), $R_{act}$ (second row), $R_{deact}$ (third row), and $R_{sed}$ (fourth row) from a box model using a Lagrangian microphysics approach for the five largest $n_{in}$: 10, 20, 30, 40, 50 $\mathrm{cm}^{-3}\mathrm{s}^{-1}$.

in Shaw et al., 2023) can still predict the scaling relationships in fast and slow microphysics regimes that are consistent with large-eddy simulations (Fig. 1) and Pi chamber experiments (Fig. 7 in Shaw et al., 2023). The nice performance of the theoretical model and the box model suggests that turbulence is not the direct factor in generating various microphysics regimes, including provoking cloud oscillation. As long as the cloud is well mixed (due to turbulence), various microphysics
regimes (e.g., slow, fast, oscillation) can occur under different aerosol injection rates (for monodisperse aerosols like in this study).

It is interesting to see that droplet deactivation can still occur even though $s$ is always positive in the box model (Fig. 11). This is also likely to be true in LES when using $\mathrm{CL_{Haze}}$ scheme, in which $s$ oscillates around $s_{crit}$ and $\sigma_s << s_{crit}$. One question is what drives droplet deactivation in a supersaturated environment. Here, droplet deactivation occurs due to the curvature effect:
Although haze particles can be activated to droplets when $s > s_{crit}$, the subsequent decrease of $s$ (like $s$ oscillation in our case) can lead to droplet deactivation when $s$ is smaller than the saturated saturation ratio over small cloud droplets (see green line in Fig.1 of Nenes et al., 2001). In addition, there is one difference in handling droplet deactivation between the $\mathrm{CL_{CCN}}$ and the $\mathrm{CL_{Haze}}$ schemes. If droplets are deactivated, they go back to the dry aerosol category when using the $\mathrm{CL_{CCN}}$ scheme. When using the $\mathrm{CL_{Haze}}$ scheme, droplets stay as haze particles which can still consume water vapor and contribute to liquid water
content. The latter has feedback in $s$ which is critical to trigger cloud oscillation that we will discuss next.

When $s < s_{crit}$, droplet activation is suppressed in the bulk region. This is true for both $\mathrm{CL_{Haze}}$ and $\mathrm{CL_{CCN}}$ schemes. However, when using the $\mathrm{CL_{Haze}}$ scheme, the contribution of haze water content to the total liquid water content increases under

this condition ($s < s_{crit}$) due to continuous aerosol injection and droplet deactivation (as discussed above). The sink of water vapor via condensational growth decreases due to the decrease of cloud droplet concentration, which can lead to an increase in $s$, considering that the source of water vapor from chamber surfaces is constant. When $s > s_{crit}$, droplet activation is active again. In contrast, when using the $CL_{CCN}$ scheme, haze water content is not considered and cloud droplet content is equivalent to liquid water content. In addition, both $s$ and $\sigma_s$ are buffered to approach 0 under polluted conditions, and there is no restoring force to increase $s$.

### 3.2.4 Cloud collapse

For the simulations above, the side walls are set to be saturated with respect to water. In reality, the side walls in the Pi chamber could be subsaturated, which could enhance droplet deactivation. To investigate the impact of side wall humidity ($RH_{wall}$) on cloud oscillation, we set $RH_{wall}$ to be 90, 70, 50, and 30% for $n_{in} = 20$ cm$^{-3}$s$^{-1}$. This is similar to the entrainment of subsaturated air into a natural cloud. Figure 12 shows the time series of domain averaged $q_l$, $N_d$, $N_a$ or $N_h$, $N_T$, and $r_d$, while Figure 13 shows the corresponding $R_{act}$, $R_{deact}$, $s$, $\sigma_s$, and $P$ (as Figs. 5 and 6). Results indicate that $q_l$ decreases with $RH_{wall}$ (Fig. 12 first row). This is because subsaturated side walls serve as a water sink to evaporate droplets and thus enhance haze-cloud interactions. Note that $q_h$ can be as large as $q_c$ (e.g., for $RH_{wall}$ of 30% at the end of the simulation), which cannot be captured when using the $CL_{CCN}$ scheme. The cloud always reaches a steady state when using the $CL_{CCN}$ scheme. In contrast, when using the $CL_{Haze}$ scheme, the cloud oscillates for $RH_{wall}$ of 90 and 70%, but it can reach a steady state for $RH_{wall}$ of 50%, and more interestingly, it tends to collapse for $RH_{wall}$ of 30% (Fig. 12 second row). Here we define "cloud collapse" as the significant decrease in $N_d$ at low $RH_{wall}$ conditions. It is also clear to see that the bulk $s$ is negative in the cloud collapse regime (fourth row in Fig. 13). Note that the cloud does not dissipate completely because $q_c$ still reaches a steady state, probably due to droplet activation near the top and bottom surfaces where the local $s$ can be still larger than $s_{crit}$ (similar to the high $s$ observed near the surface in Wang et al. (2024a).

Our results suggest that cloud oscillation and cloud collapse result from haze-cloud interactions in a homogeneous and inhomogeneous supersaturation field, respectively. When the side walls are close to be saturated, the supersaturation field is almost homogeneous everywhere in the chamber except very close to the top and bottom surfaces. Such a homogeneous supersaturation field allows synchronized droplet activation or deactivation to occur throughout the entire chamber and thus leads to cloud oscillation as explained above and Fig. 8. However, when the side walls are considerably drier, the supersaturation field in the chamber is not homogeneous: air close to the side wall is subsaturated while air close to the center, top, and bottom surfaces is supersaturated. Such an inhomogeneous field causes droplet activation in one region and deactivation in another region. For a moderate dry side wall (i.e., $RH_{wall}$ of 50%), a steady state might be reached if the net activation rate is balanced by the droplet sedimentation rate. For an extremely dry side wall (i.e., $RH_{wall}$ of 30%), the chamber can be considered as a machine to efficiently transfer cloud droplets to haze particles over time, leading to the cloud collapse.

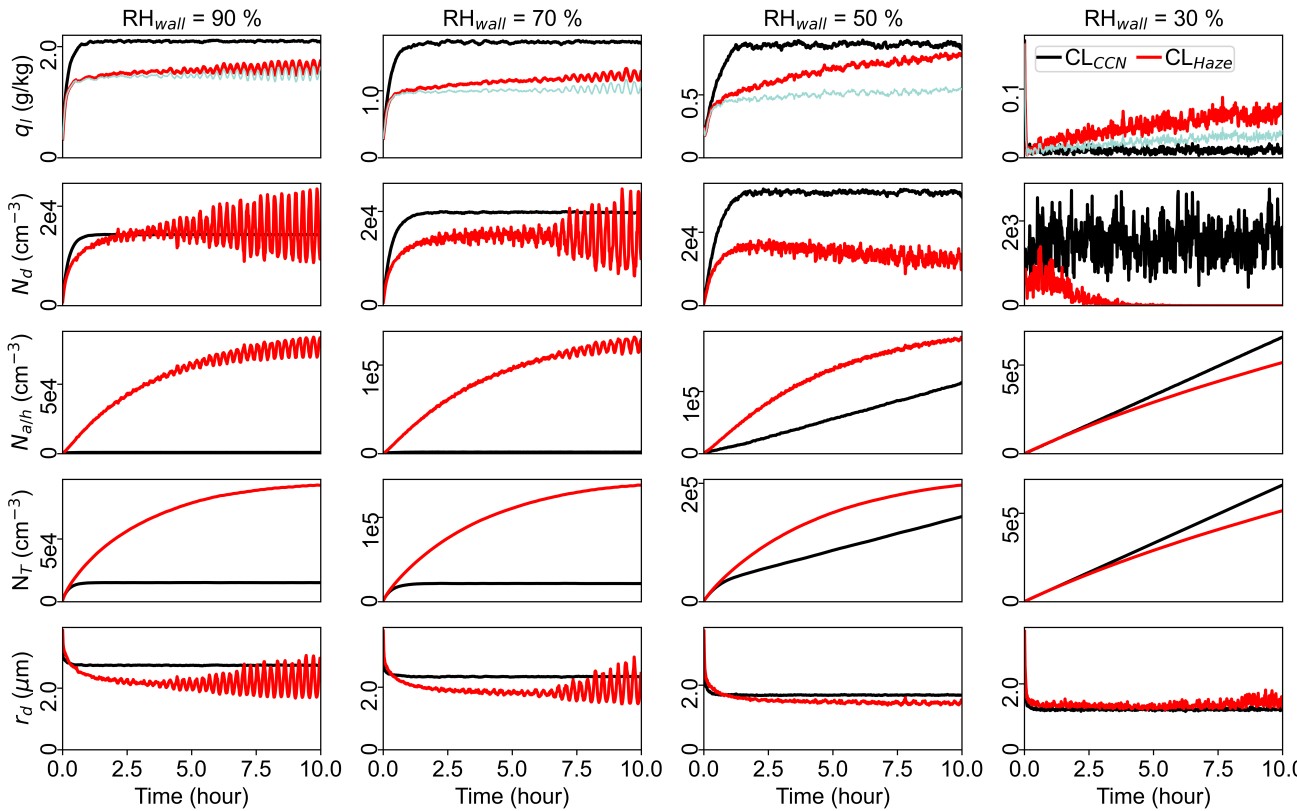

**Figure 12.** Time series of domain-averaged $q_l$ (first row), $N_d$ (second row), $N_a$ or $N_h$ (third row), $N_T$ (fourth row), and $r_d$ (fifth row) at a $n_{in}$ of 20 cm$^{-3}$s$^{-1}$ with four different size wall relative humidity, RH$_{wall}$ = 90, 70, 50, and 30%. The light blue line in the first row represents the cloud water mixing ratio ($q_c$) when using the CL$_{Haze}$ scheme.

### 3.2.5 Haze-only regime

So far, our results show that besides slow and fast microphysics regimes, there exists a cloud oscillation regime at a high aerosol injection rate due to haze-cloud interactions. In the oscillation regime, the oscillation frequency decreases and the haze number concentration increases as $n_{in}$ increases. It raises a question of what would happen if $n_{in}$ is extremely large. Would there be another regime in which there are only haze particles and no cloud droplets? Here, we develop a simple model to investigate the properties of a postulated haze-only regime.

Let us assume only haze particles exist in the chamber at an extremely high aerosol injection rate. Following the approach of Shaw et al. (2023) (Eqs. 56 and 57 therein), in the steady state, the mean air temperature would be higher than the reference temperature (i.e., $T_0$, same as in our Eq. 1) due to latent heat release from the formation of haze particles,

$$T = T_0 + \tau_m \frac{L}{c_p} \frac{dq_l}{dt}\bigg|_{diff}. \tag{7}$$

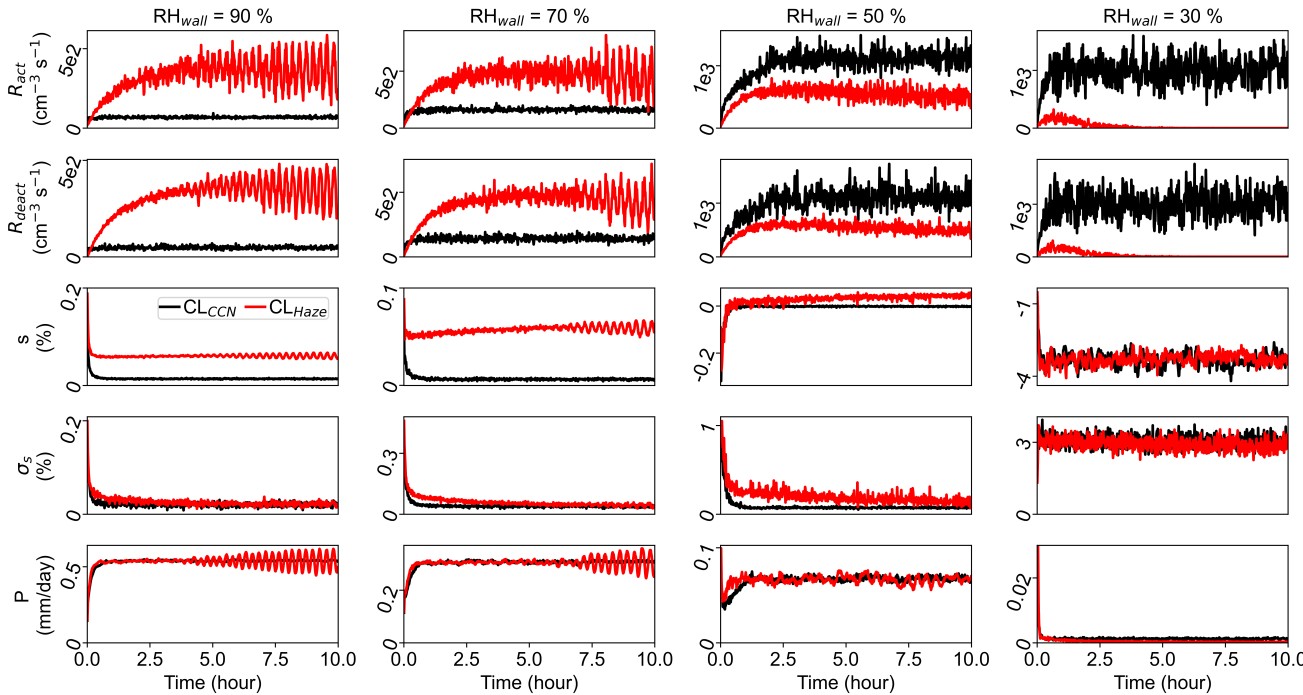

**Figure 13.** Same cases in Fig. 12, but showing time series of domain-averaged $R_{act}$ (first row), $R_{deact}$ (second row), $s$ (third row), $\sigma_s$ (fourth row), and $P$ (fifth row) at a $n_{in}$ of 20 cm$^{-3}$s$^{-1}$ with four different size wall relative humidity, RH$_{wall}$ = 90, 70, 50, and 30%.

Similarly, $q_v$ would be smaller than the reference water vapor mixing ratio (i.e., $q_{v0}$, same as in our Eq. 2) due to water uptake by haze particles,

$$q_v = q_{v0} - \tau_m \frac{dq_l}{dt}\bigg|_{diff}. \tag{8}$$

$dq_l/dt|_{diff}$ in Eqs. 7 and 8 indicates that only the contribution via diffusional growth is considered here. In the haze-only regime, condensation is dominated by the formation of haze particles,

$$\frac{dq_l}{dt}\bigg|_{diff} = \frac{4}{3}\pi\rho_l r_{eq}^3 n_{in}. \tag{9}$$

Here $r_{eq}$ is the equilibrium haze particle radius at a given $s < s_{crit}$, which depends on the environmental fractional relative humidity (RH$\equiv$1+s) and on properties of the substance. We assume that particles reach their equilibrium size within a very short time. $r_{eq}$ can be expressed as a function of RH for values near but smaller than unity based on Eq. 10 of Lewis (2019), where the constants are those for sodium chloride:

$$r_{eq} = r_{dry}\frac{1.04}{\left[1 - RH + \left(\frac{0.99\,nm}{r_{dry}}\right)^{3/2}\right]^{1/3}}. \tag{10}$$

This expression is accurate to within 5% for values of RH between 99% and 100% for dry aerosol radius ($r_{dry}$) larger than 10 nm. A similar expression of $r_{eq}$ is also derived by Khvorostyanov and Curry (2007) (Eq. 16 therein).

Meanwhile, a steady-state haze-only system requires that the formation of haze particles through injection is balanced by their loss due to sedimentation,

$$n_{in} = \frac{N_h}{\tau_{sed}}. \tag{11}$$

Here $N_h$ is the haze number concentration and $\tau_{sed}$ is the characteristic sedimentation time of haze particles with a radius of $r_{eq}$ (see Eq. 6).

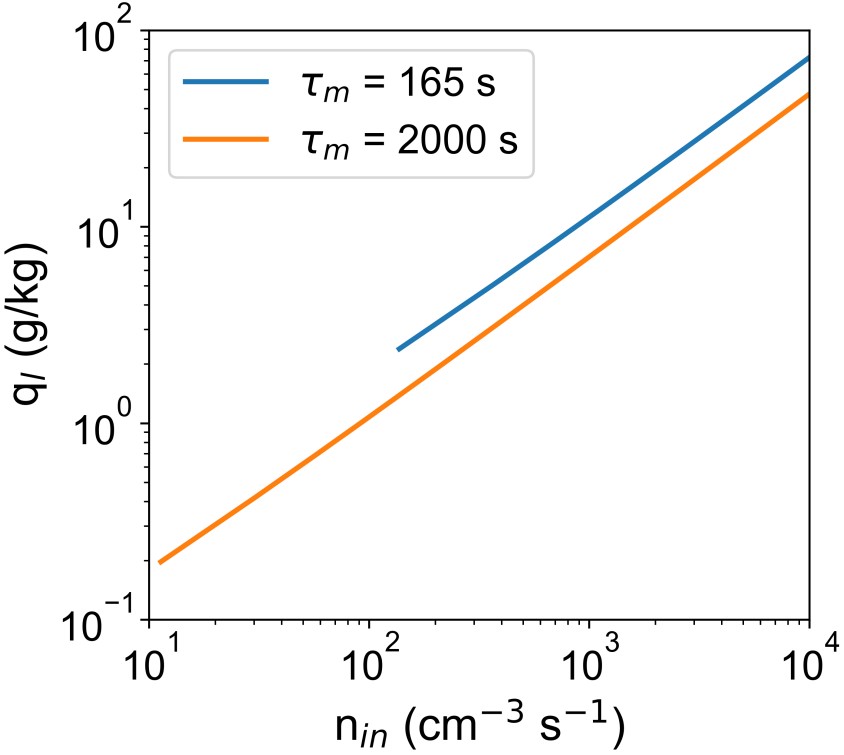

**Figure 14.** Change of equilibrium liquid water mixing ratio with $n_{in}$ in the haze-only regime. Results are calculated numerically based on the Eqs. 7 to 11, with $T_0 = 290$ K and $q_{v0} = 13.9$ g kg$^{-1}$. Blue and orange lines are for $\tau_m = 165$ s and 2000 s, respectively. The left ends of the two lines are determined at RH=100%.

For a given forcing ($T_0$, $q_{v0}$, and $\tau_m$) and aerosol ($r_{dry}$) condition, we can calculate the equilibrium liquid water mixing ratio at the haze-only steady state by solving Eqs. 7 − 11 numerically. For a direct comparison with the above results, we set $T_0 = 290$ K, $q_{v0} = 13.9$ g kg$^{-1}$, and $\tau_m = 165$ s, same as those used in the box model. Figure 14 shows that $q_l$ increases
with $n_{in}$ linearly in log-log space with a slope of about 0.83, which is steeper than that in the fast microphysics regime (0.67).

Note that we only simulate the haze-only regime in the subsaturated environment here (i.e., RH<100%, the left ends of the two lines in Fig. 14 are determined at RH=100%), and the slope should be related to the RH dependence of $r_{eq}$ (Eq. 10). Results show that the required $n_{in}$ to reach this haze-only regime is extremely high, hundreds to thousands cm$^{-3}$s$^{-1}$, and $q_l$ is also exceptionally high, tens to hundreds g kg$^{-1}$. The main reason for the high $n_{in}$ and $q_l$ is that a huge amount of slowly sedimenting haze particles are needed to balance the relatively strong forcing term to replenish water vapor so that $s < 0$ all the time. Such high $q_l$ is likely unrealistic and unachievable in the real chamber due to factors not considered in the model (see the following section). However, if $\tau_m = 2000$ s, implying a much weaker forcing, $q_l$ in the haze-only regime can be less than 1 g kg$^{-1}$ for a more realistic $n_{in}$ (Fig. 14).

So far, we have demonstrated the existence of the haze-only microphysics regime in an idealized scenario. One question is whether the haze-only regime is stable. We expect that the steady state in the haze-only regime is stable for a given $n_{in}$. This is because the aerosol injection rate should be equal to the sedimentation rate of haze particles in the steady state (see Eq. 11). If there is a positive (or negative) perturbation of $N_h$, the sedimentation rate would increase (or decrease), leading to a net decreasing (or increasing) tendency in $N_h$ for a given $n_{in}$. This feedback is trying to bring $N_h$ back to its steady state value. Of course, this is only our conjecture, not a formal proof. Further efforts are needed to understand the onset of oscillation, the transition between oscillation regime and haze-only regime, and the stability of the haze-only regime.

### 3.2.6 Impact of a haze sink

So far, the only sink for aerosol particles is activation. At high aerosol injection rates, activation is suppressed, and thus, they can accumulate when using the CL$_{\text{CCN}}$ scheme (see black lines in Figs. 5 and 12, third row). Similarly, the sink for haze particles is dominated by activation because their sedimentation speed is very small. We have shown that a chamber with subsaturated side walls can efficiently transfer cloud droplets to haze particles over time, leading to haze accumulation when using the CL$_{\text{Haze}}$ scheme (red line in Fig. 12, third row). In reality, these unactivated particles (aerosols or haze particles) can also be lost by side walls, coagulation, sedimentation, or droplet scavenging, preventing their concentration from approaching infinity.

To investigate the impact of the sink of haze particles on cloud properties, especially in the cloud oscillation regime, following Thomas et al. (2023) and Wang et al. (2024c) (Eq. 1 therein), a wall-loss timescale ($t_{wl}$) is applied to constrain $N_h$ when using the CL$_{\text{Haze}}$ scheme as

$$\delta N_h = -N_h \frac{\Delta t}{t_{wl}}. \tag{12}$$

Here $\Delta t$ is the time step of the simulation, and $\delta N_h$ is the loss of haze particles due to walls after each time step. $t_{wl}$ is one over the particle rate loss coefficient ($\beta$) due to the walls. $\beta$ can be estimated from the deposition velocity ($v_{dep}$) and wall area ($A$) to volume $V$ ratio (for the Pi chamber $A/V = 4$ m$^{-1}$): $\beta = v_{dep}A/V$. For simplification (i.e., neglecting the impact of other factors, such as particle size and turbulence, on $v_{dep}$), we set $v_{dep} = 10^{-4}$ m s$^{-1}$, a typical value for the deposition velocity for particles with a diameter of 2 $\mu$m (see fig. 4 in Lai, 2002), which give us $\beta = 4 \times 10^{-4}$ s$^{-1}$ or $t_{wl} = 2500$ s. Results show that oscillation still exists for $n_{in} \geq 20$ cm$^{-3}$s$^{-1}$, but with a smaller amplitude (red line in Fig. 15). The oscillation frequency

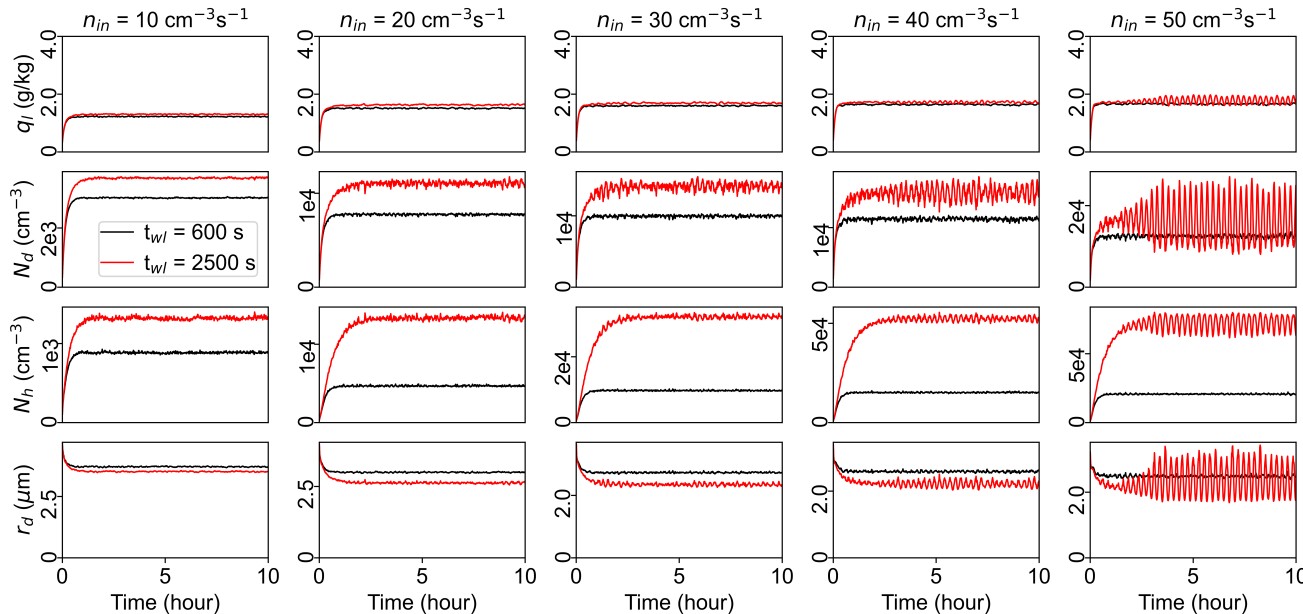

**Figure 15.** Similar to Fig. 5, time series of domain-averaged $q_l$ (first row), $N_d$ (second row), $N_h$ (third row), and $r_d$ (fourth row) for five different $n_{in}$: 10, 20, 30, 40, and 50 cm$^{-3}$s$^{-1}$, but considering the loss of haze particles due to the side wall. Different line colors represent simulations using different wall-loss timescales ($t_{wl}$ in Eq. 12): $t_{wl} = 600$ s (black) and 2500 s (red).

is also higher than before (compare Figs. 5 and 15 for the same $n_{in}$). Although we only consider the loss of haze particles

due to walls here, there are some other types of haze sinks, such as Brownian coagulation (Baker and Charlson, 1990) and scavenging (Sellegri et al., 2003), which might lead to a smaller effective $t_{wl}$ in the real chamber. For another sensitivity test, we set $t_{wl} = 600$ s, the same value Thomas et al. (2019) used to constrain particle concentration for the Pi chamber simulation. Results show that the oscillation is barely seen (black line in Fig. 15). Also note that $N_h$ increases with $n_{in}$, but its value is one order of magnitude smaller than before (Fig. 15 vs. 5, third row). Combined with the "cloud collapse" findings, our results

suggest that achieving a high concentration of haze particles and synchronized activation throughout the chamber are two key factors for the cloud to stay in the oscillation regime.

## 4   Conclusions and discussion

In this study, we conducted a series of large-eddy simulations of the Pi chamber using a haze-capable bin microphysics scheme (CL$_{\text{Haze}}$) developed by Yang et al. (2023) to explore haze-cloud interactions over a wide range of aerosol injection rates

(0.001 cm$^{-3}$s$^{-1}$ $\leq n_{in} \leq$ 50 cm$^{-3}$s$^{-1}$). Results are compared with simulations using a CCN-based bin microphysics scheme (CL$_{\text{CCN}}$). The CL$_{\text{CCN}}$ scheme adopts a Twomey-type activation parameterization which is widely used in atmospheric cloud simulations, while the CL$_{\text{Haze}}$ scheme can properly resolve the growth of haze particles and the activation process. Our ob-

jectives were to investigate (1) the influence of different aerosol injection rates on cloud properties, and (2) the importance of haze-cloud interactions in a convection cloud chamber as well as in analogous natural cloud systems. For objective 1, we especially focused on the impact of $n_{in}$ on cloud droplet number concentration ($N_d$), liquid water mixing ratio ($q_l$), and droplet size distribution, and compared results with previous analytical studies (Krueger, 2020; Shaw et al., 2023). Objective 2 is motivated by Yang et al. (2023) showing that cloud microphysical properties gained with the $CL_{CCN}$ scheme are similar to those using the $CL_{Haze}$ scheme, raising the question of whether we need to consider haze-cloud interactions. However, only two aerosol injection rates were investigated in Yang et al. (2023). Here, we explored the consistency of the $CL_{CCN}$ scheme and the $CL_{Haze}$ scheme over a wider range of aerosol injection rates. Low-dimensional models are also employed to explore the impact of $n_{in}$ on cloud properties. In short, we confirm slow and fast microphysics regimes reported in previous studies (Shaw et al., 2023). We also find new microphysical regimes at high aerosol injection rates, cloud oscillation and haze-only, as illustrated in Fig. 16.

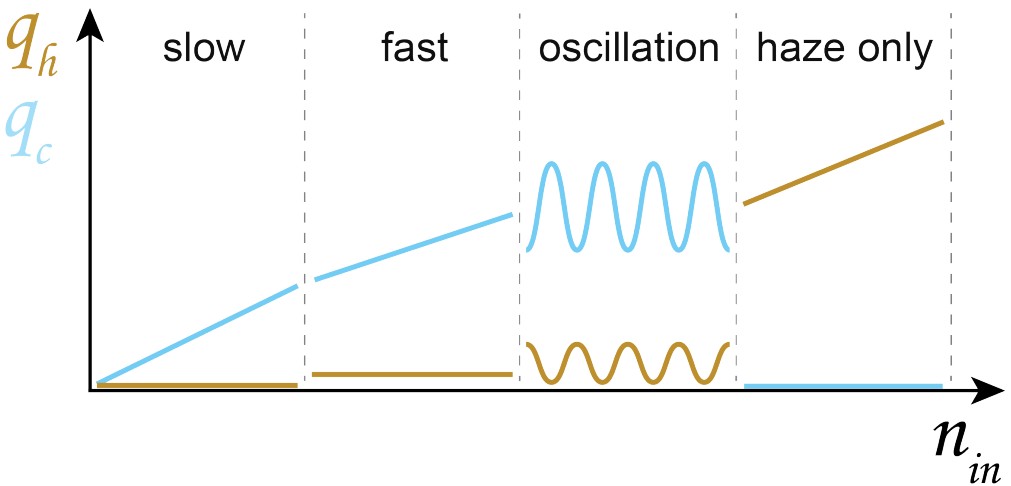

**Figure 16.** A schematic illustration of $q_c$ or $q_h$ and $n_{in}$ relationships in different microphysics regimes: slow, fast, oscillation, and haze-only.

Slow and fast microphysics regimes were observed at small and moderate aerosol injection rates, respectively. The change of cloud properties with aerosol injection rate in these two regimes agreed with previous analytical studies (Chandrakar et al., 2020; Shaw et al., 2023). Specifically, for small aerosol injection rates ($n_{in} < 0.1$ cm$^{-3}$s$^{-1}$), the cloud was in the slow microphysics regime where droplets grow at a high supersaturation before they fall out, leading to a linear relationship between $N_d$ and $n_{in}$ as well as $q_l$ and $n_{in}$. For moderate aerosol injection rates ($0.1$ cm$^{-3}$s$^{-1} \leq n_{in} \leq 10$ cm$^{-3}$s$^{-1}$), the cloud was in the fast microphysics regime with $N_d \sim n_{in}^{5/3}$ and $q_l \sim n_{in}^{2/3}$, consistent with the theoretical prediction in Shaw et al. (2023). In addition, droplet size distributions in the steady state became narrower and shifted to smaller sizes due to the increase in $n_{in}$, and the shape of the distribution also agreed reasonably well with analytical estimates (Chandrakar et al., 2020; Liu and

Hallett, 1998; Krueger, 2020). But those analytical estimates do not capture the distribution properties at large $n_{in}$ where haze mode is present.

The most striking phenomena are cloud oscillation, cloud collapse, and haze-only regimes occurs at high aerosol injection rates when using the $CL_{Haze}$ scheme. In contrast, cloud always reaches a steady state when using the $CL_{CCN}$ scheme. Haze-cloud interactions are responsible for the occurrence of these microphysics regimes. Specifically, in the cloud oscillation regime, $s$ oscillates around $s_{crit}$ and $\sigma_s \ll s_{crit}$. Under this condition, the cloud system is buffered by a huge amount of haze particles and cloud droplets. Droplet activation is controlled by the mean supersaturation rather than supersaturation fluctuation. Droplet deactivation can still occur in supersaturated environment ($0 < s < s_{crit}$) due to the curvature effect. The oscillation of $s$ around $s_{crit}$ leads to the oscillation of droplet activation and deactivation, and further causes the oscillation of cloud properties. In a chamber with relatively humid side walls, the supersaturation is more homogeneous in the chamber, and droplets at different locations experience similar supersaturation, leading to synchronized activation ($s > s_{crit}$) of a huge number of droplets across the whole chamber – the main reason for cloud oscillation. In contrast, cloud collapse occurs when the side walls are relatively dry. Under this condition, supersaturation in the chamber is more inhomogeneous: droplets close to the side walls tend to be deactivated to haze particles while droplets away from the side walls tend to grow. The separation of droplet activation (in regions near the center, top, and bottom surfaces) and deactivation (in regions near the side walls) make the chamber an efficient machine to transfer cloud droplets to haze particles – the fundamental reason for cloud collapse. The haze-only regime occurred at extremely high aerosol injection rates. In this regime, $s$ is much smaller than $s_{crit}$, and it can be negative (corresponding to RH<100%). Droplet activation is suppressed and the formation of haze particles is balanced by their loss due to sedimentation.

In the real chamber, haze particles can also be removed through other mechanisms, such as wall loss and scavenging, which could constrain the haze number concentration. Therefore, clouds might struggle to achieve oscillation and haze-only regimes, especially when the source term to maintain high supersaturation is strong, e.g., a large temperature difference between top and bottom surfaces, like in this study. Haze-cloud oscillation is more likely to occur under conditions of weak supersaturation forcing, e.g., a small temperature difference between top and bottom surfaces in a convection chamber, or small updraft velocity in the real atmosphere. Recently, Gutiérrez et al. (2024) solved coupled equations for droplet growth and supersaturation development in a rising cloud parcel. Their analysis also predicts the oscillation between haze and cloud droplets under certain conditions, e.g., low air vertical velocity and high aerosol number concentration. The fundamental reason for cloud oscillation stems from the non-linear interactions in the coupled haze-cloud-supersaturation system (Arabas and Shima, 2017). Such a system is analogous to other predator-prey systems observed in nature, which causes similar oscillation behaviors, such as oscillation in open-cellular convection or in aerosol-cloud-precipitation system (Koren and Feingold, 2011). However, it should be mentioned that cloud oscillation reported in Gutiérrez et al. (2024) is not the same as oscillation reported in this study: they only have one size of droplet/haze that varies in time, while we have coexisting haze and cloud droplets.

Our results suggest that haze-cloud interactions are very important when air supersaturation is close to the critical supersaturation of aerosols. This condition happens in the Pi chamber at high aerosol injection rates, as shown in this study, and it can also occur in the atmosphere, for example, when cloud or fog is close to the source of intense natural and anthropogenic aerosol emissions. Studies have shown the possibility of fog consisting of just unactivated haze particles in highly polluted

environment (e.g., Klemm et al., 2016). The unactivated haze particles can significantly impact fog optical properties, such as visibility and radiation (Boutle et al., 2018), as well as cloud optical properties, i.e., cloud albedo (Hoffmann et al., 2022). Proper simulation of haze-cloud interactions requires resolving haze particles as well as the associated droplet activation and

555 deactivation processes, rather than relying on Twomey-type activation parameterization. Also note that monodisperse aerosol with a dry radius of 62.5 nm is used in this study. We expect haze-cloud interaction might be more important for larger aerosol particles because their critical supersaturation gets smaller, their equilibrium wet radius gets larger, and the activation/deactivation time scale could get longer (Hoffmann, 2016). In addition, aerosol particles in nature vary in size and composition, and haze-cloud interactions might be more important for polydisperse aerosols (see Fig. 5 in Richter et al., 2021), which is also

worth exploring in the future.

*Code availability.* The SAM model was kindly provided by Prof. Marat Khairoutdinov of Stony Brook University and is publicly available at http://rossby.msrc.sunysb.edu/ marat/SAM.html.

*Data availability.* Data and Python Code for figure generation are available at https://doi.org/10.5281/zenodo.14002522.

*Author contributions.* F.Y., R.A.S., and F.H.: conceptualization. F.Y., H.F.S., and A.W.: code development and debug. F.Y. and H.F.S.: con-

565 ducting simulations. F.Y. and P.H.: data analysis and visualization. F.Y.: original draft preparation. H.F.S., R.A.S., F.H., P.H., A.W., and M.O., writing - review and editing.

*Competing interests.* The contact author has declared that none of the authors has any competing interests.

*Acknowledgements.* We thank Prof. Shin-ichiro Shima and the other reviewer for their valuable comments and suggestions.This work was supported by Office of Science Biological and Environmental Research program as part of the Atmospheric Systems Research program.

Brookhaven National Laboratory is operated by Battelle for the U.S. Department of Energy under contract DE-SC0012704. PNNL is operated for the Department of Energy by Battelle Memorial Institute under Contract DE-AC05-76 RL01830. H. Fahandezh and R.A. Shaw were supported by NSF grant AGS-2133229. F.Hoffmann is supported by the German Research Foundation (DFG) under grant HO 6588/1-1. We thank Subin Thomas, who was involved with early large eddy simulations that hinted at the possibility of cloud oscillations. We thank Ernie Lewis for helpful discussions that made us aware of Equation 10. F. Yang also thanks Kamal Kant Chandrakar and Silvio Schmalfuß for

helpful discussions.

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
