# Peer review of "Microphysics regimes due to haze-cloud interactions: cloud oscillation and cloud collapse"

_EGUsphere, 2024_

## Referee Comment (RC2)

**Review of "Microphysics regimes due to haze-cloud interactions: cloud oscillation and cloud collapse" by Fan Yang et al.**

This manuscript could be considered for publication after a major revision.

The authors conducted a series of large-eddy simulations (LESs) of a cloud in a convection chamber using a haze-capable Eulerian-based bin microphysics scheme to explore haze-cloud interactions over a wide range of aerosol injection rates. They observed three microphysics regimes as they increased the aerosol injection rate: slow microphysics regime, fast microphysics regime, and cloud oscillation. The cloud oscillation is a new phenomenon being reported for the first time in this study. To understand the physical mechanism, they conducted a detailed analysis by introducing a box model. They also found that cloud collapse can occur if the side wall humidity is low. By solving a haze-only box model analytically, they also found the existence of a haze-only solution when aerosol injection rate is high.

As the LES using Twomey-type activation reproduced the slow and fast microphysics regimes but could not capture the cloud oscillation, they concluded that the haze-cloud interaction is critical in polluted conditions, but we could still use Twomey-type activation parameterizations for less polluted conditions.

The manuscript is relatively well written. However, the analysis is not comprehensive enough, and this makes the author's main conclusions not fully convincing. In particular, I see two major issues in this study:
1. They concluded that we could still use Twomey-type activation parameterizations for less polluted conditions. But, in my opinion, the conditions they tested are rather limited and the conclusion could be misleading.
2. I was confused that the author's interpretation regarding the cloud oscillation is sometimes not consistent with the plots they presented. A more careful analysis on supersaturation, supersaturation fluctuation, activation rate, deactivation rate, and sedimentation rate is desired. I believe it should foster a deeper understanding about the cloud oscillation phenomenon.

Please also see my more detailed comments provided below.

I believe the quality of the study will be significantly enhanced if these points are addressed. I look forward to reading the revision of this manuscript.

**Major Comments**

1) **[request] P. 10 ll. 207-208 "The apparent transition between slow and fast regimes as shown in Fig. 1 provides an opportunity to estimate $\tau_m$"**
   From the discussion on p. 16, it seems the authors are thinking that the transition point is $n_{in} = 0.1\,\mathrm{cm}^{-3}\mathrm{s}^{-1}$ and $\tau_m \approx 70\,\mathrm{s}$, but it is indicated only implicitly on p. 10. Please clarify this point.

Well, it should be possible to calculate the turbulent mixing time from the flow field directly, e.g., by calculating the integral time scale of the turbulence.

2) **[request] Figure 4**

Please also show the time evolution of total particle number $(N_d + N_{a/h})$, activation rate, deactivation rate, sedimentation rate, $s$, and $\sigma(s)$. This should provide more insight into understanding the mechanism.

3) **[request] P. 11 l. 220 "Note that Na increases with time for nin ≥ 10 cm−3s−1."**

Do you mean $n_{in} \geq 40\,\mathrm{cm}^{-3}\mathrm{s}^{-1}$? Please clarify.

4) **[suggestion] P. 11 l. 224 "The oscillation period increases as nin increases, ..."**

It should be also informative to point out that the oscillation amplitude increases as $n_{in}$ increases.

5) **[request] Figure 5i**

For a better comparison, please use the same color bar as in Figure 6.

6) **[question] P. 11 ll. 234–235 "The sharp increase in Nd (Fig. 5b) corresponds to a larger activation rate (Fig. 5c) due to the enhanced supersaturation (Fig. 5i), …"**

From Figs. 5c and 5f, it looks like the mean $R_{deact}$ is always larger than mean $R_{act}$, though on average $R_{act} - R_{deact} = n_{in}$ has to be satisfied. Why is it?

7) **[question] P. 11 l. 235 "..., while the decrease in Nd corresponds to a larger deactivation rate and a smaller supersaturation."**

If we compare Figs. 5f and 5i, the deactivation rate is larger even when supersaturation is larger, which is counterintuitive. Is this due to supersaturation fluctuation?
On a related note, is removal by sedimentation much smaller than $R_{deact}$ and $R_{act}$?

8) **[question] P. 11 l. 240 "When s > scrit (scrit ≈ 8‰ in this study), ..."**

From Fig. 5i, it looks like $s < s_{crit}$ always holds. Why is it?

9) **[question] P. 12 ll. 243–245 "Shortly thereafter, Nd decreases because droplet activation is suppressed when s < scrit, and meanwhile, droplets are lost due to sedimentation and deactivation."**

Which is dominant, sedimentation or deactivation? From Figs. 5i and 6, mean supersaturation $s$ is always positive. Then, does deactivation occur due to supersaturation fluctuation? How big is $\sigma(s)$?

10) **[question] P. 12 l. 249 "It is interesting to see that the oscillation evolves with time clockwise in qh −Nh diagram (Fig. 6c) …"**

If we compare the color and size of the circles in Figs. 6c and 6d, it looks like the sequence in Fig. 6c is in the opposite order, i.e., anticlockwise.

11) **[comment] Pp. 12–13 ll .236–250**

From the observation described in this section, it is not clear why cloud oscillation does not occur when using the CL_CCN scheme.

**12) [suggestion] Eqs. 1, 2, and 4**

The $dq_l/dt$ here represents the change of $q_l$ through diffusional growth, but droplet removal by sedimentation also changes $q_l$. To avoid confusion, the authors should clarify this point, e.g., by using a subscript: $dq_l/dt|_{diff}$.

**13) [question] Eq. 3**

If I understand correctly, the supersaturation fluctuation is not considered in this box model, right? Is it not important for the phenomenon?

**14) [question] P. 16 l. 299 "T0 and qv0 are set to be 290 K and 13.9 g kg−1, ..."**

What is the corresponding supersaturation $s_0$?

**15) [request] P.16 l.302 "... the estimated τm for Da = 1 based on LES results ..."**

Please clarify that the estimated $\tau_m$ was $70\,\mathrm{s}$.

**16) [request] Figure 8**

Please also show the time evolution of total particle number $(N_d + N_h)$, activation rate, deactivation rate, sedimentation rate, and $s$.

**17) [question] P.16 l.316 "In contrast, simulations using the CLCCN scheme do not show oscillations ..."**

What will happen if we use Twomey activation for this box model? Fig. 4 revealed that LES with CL_CCN does not show oscillation, but it seems to me nothing prevents the Twomey box model from exhibiting oscillation?

**18) [question] Sec. 3.2.2 "Cloud oscillation in a box model"**

Additional questions about this section:
- Because supersaturation fluctuation is not considered in this model, $s > 0$ always holds, and deactivation of droplets does not occur in this model. Is this correct?
- Instead, the droplets are removed from the system only by sedimentation in the box model. Is this correct?
- The oscillation amplitude of $N_d$ in Fig. 8 is smaller than that in Fig. 4. Is this because of the absence of deactivation in the box model?
- Because supersaturation fluctuation is not considered in this model, almost all haze particles should be activated when $s > s_{crit}$. This is the reason why $N_h$ decreases to almost 0 in Fig. 8 (though this is not happening in Fig. 4). Is this correct?

**19) [question] P.18 l.344 "..., the oscillation frequency approaches zero)?**

Why do you think the frequency approaches zero when switching to the haze-only regime?

In dynamical systems theory, there are various types of bifurcations that are responsible for the onset of oscillation. If the frequency approaches zero, it suggests it is an "infinite period bifurcation". On the other hand, if the oscillatory solution arises when a fixed point (haze-only solution) is destabilized, it is a Hopf bifurcation. Then, the frequency is finite at the bifurcation point. See, e.g., Strogatz (2014).

**20) [suggestion] Eqs. 7–11**

Again, it should be clarified that only the contribution via diffusional growth is considered, e.g., by using a subscript: $dq_l/dt|_{diff}$.

In particular, Eq. 11 is confusing if $dq_l/dt$ is used on the l.h.s.; if we take it literally, it indicates $q_l$ grows exponentially! But, of course, the correct meaning is $dq_l/dt|_{diff} = q_l/\tau_{sed}$. I would suggest the expression $n_{in} = N_h/\tau_{sed}$ as an equivalent but more intuitive alternative.

**21) [question] Eq. 10**

I think the use of this formula is not appropriate for this analysis, because $r_{eq} \to \infty$ as $RH \to 1$. How about simply assuming $r_{eq} = \text{const.}$?

**22) [question] Figure 10**

How are the left ends of the two lines determined? Are they corresponding to $RH = 1$? If so, Eq. 10 should not be used because it is not accurate when $RH$ approaches 1.

**23) [request] P.21 l.374 "..., so that s < scrit all the time."**

Because of the use of Eq. 10, the analysis is valid only for $s < 0 (< s_{crit})$. Please clarify this point.

**24) [comment] Sec. 3.2.4 "Haze-only regime"**

The existence of the haze-only solution is presented in this section, but it does not guarantee it is stable.

**25) [request] P.23 l.433 "deactivation (s < scrit)"**

Do you mean $s < 0$?
In both CL_haze and CL_CCN, deactivation occurs only when the supersaturation is locally smaller than zero. Hence, if there is no fluctuation, $0 < s < s_{crit}$ ($s$ represents mean supersaturation) does not induce any deactivation. (More precisely speaking, if all the activated droplets are large enough.) If the supersaturation fluctuation is large, deactivation can occur locally even when $s_{crit} < s$.

**26) [request] P.24 ll.443–444 "Haze-cloud oscillation is more likely to occur under conditions of weak supersaturation forcing, …"**

Why do you think so? Please elaborate.

**27) [comment] P.24 l.455 "Our results suggest that haze-cloud interactions are very important especially in polluted conditions."**

The authors suggest that we could still use Twomey-type activation parameterizations for less polluted conditions, but I am not fully convinced. NaCl aerosol particles with a dry radius of 62.5 nm are considered in this study, but they are relatively small. I think the haze-cloud interaction should be more important for larger aerosol particles because the equilibrium wet radius gets larger and the activation/deactivation time scale gets longer. In addition, aerosols considered are monodisperse in this study, but I also think that haze-cloud interaction is more important for polydisperse aerosols. Please see, e.g., Fig.5 of Richter et al. (2021).

These limitations of this study should be discussed and emphasized more.

**Minor Comments**

**28) [request] P.2 ll.54–55 "Shaw et al. (2023)"**

Because polydisperse aerosol injection is discussed in the previous sentence, please clarify that monodisperse aerosol injection was assumed in Shaw et al. (2023).

**29) [request] P.7 Fig. 1**

Please also show the standard deviation of supersaturation $\sigma(s)$.

What is the definition of droplet mean radius $r_d$ and droplet number concentration $N_d$ for the CL_Haze scheme? Are the haze droplets included in the population? Please clarify.

**30) [comment] P.9 Table 2 and elsewhere "Na/Nh"**

This looks like "Na divided by Nh" and I believe it is confusing to the readers. How about simply writing "Na, Nh" or "Na or Nh"?

**31) [comment] P.9 Table 2**

Please consider including units in the headers to enhance readability.

**32) [question] P.10 ll.198–199 "Note that the net activation rate (Ract−Rdeact) is close to nin for each case suggesting that the cloud reaches a quasi-steady state."**

In a quasi-steady state, the removal rate by sedimentation should be equal to the net activation rate and injection rate. Did you confirm this? To put it another way, how did you confirm that one hour is sufficient to reach a quasi-steady state?

**33) [comment] P.15 l.269 "Each particle represents numerous real particles per unit volume. We refer to this as multiplicity, …"**

Note that multiplicity is defined differently in Shima et al. (2009).

**34) [request] P.23 l.426 "But they do not capture the distribution properties …"**

Please clarify that "they" represents "analytical estimates".

**Typo**

**35) P.3 l.59 response -> respond**

**36) Eq. 4**

$G$ is not needed?

**37) Eq. 5**

If $\delta n_i$ is representing the decreased amount of multiplicity, Eq. 5 has to be

$$\delta n_i = n_i \left( 1 - \exp \left[ -\frac{\delta t}{\tau_{sed}(r_i)} \right] \right) \approx n_i \frac{\delta t}{\tau_{sed}(r_i)}.$$

**38) P.21 l.385 "following by Thomas et al." -> "following Thomas et al."**

**39) P.22 Eq. 12**

This must be $\delta N_h = N_h \Delta t / t_{ql}$.

**References**

Richter, D.H., MacMillan, T. & Wainwright, C. A Lagrangian Cloud Model for the Study of Marine Fog. Boundary-Layer Meteorol 181, 523–542 (2021). https://doi.org/10.1007/s10546-020-00595-w

S. Shima, K. Kusano, A. Kawano, T. Sugiyama, and S. Kawahara, Q. J. R. Meteorol. Soc. 135, pp.1307-1320 (2009). DOI: http://dx.doi.org/10.1002/qj.441

Strogatz, S.: Nonlinear Dynamics and Chaos: With Applications to Physics, Biology, Chemistry, and Engineering, 2nd Edn., 513 pp., Westview, 2014.

---

## Author Comment (AC1)

We appreciate the reviewers' time and effort spent on our manuscript. We are happy that the reviewer finds our paper "well motivated, written and the results illustrate novel delicate features of cloud droplet activations". We also agree with both reviewers that the physical origin of cloud oscillation was not clearly presented in the initial submitted manuscript. We have conducted more simulations, data analysis, and detailed discussion to strength our manuscript based on the reviewers' comments and suggestions. We have addressed the reviewer's comments point-by-point below without an exception. The reviewer's comments are in **blue**, our responses are in **black**, and quote from the manuscript are in ***italics*** with the corresponding changes in **bold** text.

Reviewer 1:

The present manuscript explores the interaction between haze particles and activated cloud droplets, and its effect on droplet size distribution of steady-state conditions obtained from the Pi convection chamber. The analyses are carried out by using large-eddy simulations (LES) of the Pi-chamber, with Eulerian bin microphysics that resolve the continuous process of haze growth and activation/deactivation. The main parameter of discussion is the aerosol injection rate, which regulates the amount of haze and cloud droplets in the chamber driving the system to two distinct regimes reported in a previous paper: fast and slow microphysics. It is argued that in the case of slow microphysics, corresponding to low aerosols injection rates, the transition form cloud droplets to haze is negligible, whereas a fast microphysical regime can lead to two novel regimes reported here: the cloud oscillation and cloud collapse. Oscillations are claimed to occur due to successive activation and deactivation events, whereas cloud collapse occurs when the condensation onto the particles surface is extremely efficient in reducing the saturation rate to equilibrium. The LES experiments done here clearly illustrate the importance of resolving the continuous growth of haze and its activation. Not only does it present a more accurate description of droplet formation, but it reveals new dynamical regimes.

This paper is well motivated, written and the results illustrate novel delicate features of cloud droplet activation through LES modelling. I recommend publishing this work subject to minor modifications and after the clarification of the comments below.

We appreciate the reviewer's time and effort spent on our manuscript. We have addressed the reviewer's comments point-by-point below.

General questions

1. It is mentioned on several occasions that the convection chamber is a turbulent domain. What is the role of turbulence in provoking oscillations? How is turbulence included in the box model? What fields does it affect?

We apologize that we have not made the role of turbulence in various microphysics regimes clear. The short answer is that the turbulence is not the direct factor in provoking oscillation. In the Pi chamber, turbulence helps to maintain supersaturation in the Pi chamber, keep the cloud well mixed, and generate supersaturation fluctuations that can affect droplet growth/evaporation at the microscale. From the macroscopic point of view, as long as the cloud is well mixed (due to turbulence), various microphysics regimes (slow, fast, oscillation, collapse) can occur under different aerosol injection rates (for monodisperse aerosols like in this study). This is why even the theoretical model developed in Shaw et al. (2023) does not consider turbulence, it can still predict the scaling relationships in fast and slow microphysics regimes, which are consistent with numerical simulations (Figure 1 in this study) and experiments (Figure 7 in Shaw et al., 2023). Similarly, the box model does not include turbulence but can also reproduce various microphysics regimes (including cloud oscillation). The nice performance of the theoretical model and the box model suggest that turbulence is not the direct factor to provoke oscillation.

We now add more sentences in the manuscript to discuss the role of turbulence and the origin of cloud oscillation (addressed in the following comment):

"***Additionally, $\sigma_s$ decreases with $n_{in}$ and approaches 0 due to the buffering effect of cloud droplets under polluted conditions (Fig.6 fourth row). This suggests that droplet activation is controlled by the mean supersaturation instead of supersaturation fluctuation. This is why even though turbulence is not considered, the box model (Fig. 9) and the theoretical model (developed in Shaw et al., 2023) can still predict the scaling relationships in fast and slow microphysics regimes that are consistent with large-eddy simulations (Fig. 1) and Pi chamber experiments (Fig. 7 in Shaw et al., 2023). The nice performance of the theoretical model and the box model suggests that turbulence is not the direct factor in generating various microphysics regimes, including provoking cloud oscillation. As long as the cloud is well mixed (due to turbulence), various microphysics regimes (e.g., slow, fast, oscillation) can occur under different aerosol injection rates (for monodisperse aerosols like in this study).***"

2. The origin of cloud oscillations needs to be better explained. The authors clearly explain how polluted conditions, i.e. fast microphysics regime, lead to more delicate interplay between haze and droplets. This makes as the interplay between the curvature and solute effects becomes more delicate. In the abstract, it is mentioned that "cloud oscillation arises from complex interactions between haze and cloud droplets in a turbulent cloud", however the read-outs of figure 4 (last row) show that oscillations occur at what seems the activated radius-domain. Can the authors clarify this? It would be helpful on this regard to indicate more clearly (quantitatively) when the authors consider particles to be haze, perhaps by indicating the critical Köhler radius. Do the authors observe oscillations between haze and activated droplets? The oscillations in figures 1 and 4 seem to only be occurring at radii larger that the critical radius.

This comment is also related to the previous comment. Cloud oscillation is one of the key findings of this study, and we apologize we did not explain clearly its origin in the initial submitted manuscript. We first address the reviewer's question and then explain the origin of cloud oscillation below.

First of all, the domain-averaged radius shown in Figure 4 (last raw) is calculated only from cloud droplets whose radii are larger than 1 μm, therefore the value is always in the activated radius-domain. In this study, we inject monodisperse sodium chloride aerosol with a dry radius of 62.5 nm and the corresponding critical radius of 0.92 μm. We define haze/cloud droplets as droplets whose radii smaller/larger than 1 μm.

The reviewer asks about "do the authors observe oscillations between haze and activated droplets?". The answer is yes. We show the oscillations of bulk statistical properties of haze and cloud droplets in the Pi chamber. One can clearly see the oscillation of many variables in Figures 4-6, such as haze concentration, cloud droplet concentration, cloud water content, activation rate, and deactivation rate. Haze oscillation and cloud oscillation are closely related: the increase of cloud droplet concentration is strongly correlated with the decrease of haze particle concentration through activation (see Figure 6a). This can be referred to as the oscillation between haze and cloud (the reviewer's question) in the chamber from a systematic view. However, this oscillation does NOT mean that all droplets will behave in a synchronized manner, e.g., all droplets are activated or deactivated spontaneously. The domain-averaged droplet radius in Figure 4 (last raw) shows the oscillation behavior of a well-mixed cloud system, and it does not represent the oscillation of a single cloud droplet (explained in the previous paragraph). The rapid decrease in the mean radius (associated with the rapid increase in droplet concentration) is due to the burst of newly formed small cloud droplets through activation (which shifts the mean size to a smaller value), rather than the size reduction of the existing droplets via evaporation.

[revised manuscript text omitted]

*small cloud droplets (see green line in Fig. 1 of Nenes et al., 2001). In addition, there is one difference in handling droplet deactivation between the $CL_{CCN}$ and the $CL_{Haze}$ schemes. If droplets are deactivated, they go back to the dry aerosol category when using the $CL_{CCN}$ scheme. When using the $CL_{Haze}$ scheme, however, droplets stay as haze particles which can still consume water vapor and contribute to liquid water content. The latter has feedback in s which is critical to trigger cloud oscillation that we will discuss next.*

*When s<$s_{crit}$, droplet activation is suppressed in the bulk region. This is true for both $CL_{Haze}$ and $CL_{CCN}$ schemes. However, when using the $CL_{Haze}$ scheme, the contribution of haze water content to the total liquid water content increases under this condition (s<$s_{crit}$) due to continuous aerosol injection and droplet deactivation (as discussed above). The sink of water vapor via condensational growth decreases due to the decrease of cloud droplet concentration, which can lead to an increase in s, considering that the source of water vapor from chamber surfaces is constant. When s>$s_{crit}$, droplet activation is active again. In contrast, when using the $CL_{CCN}$ scheme, haze water content is not considered, and cloud droplet content is equivalent to liquid water content. In addition, both s and $\sigma_s$ are buffered to approach 0 under polluted conditions, and there is no restoring force to increase s.*"

Specific questions

1. Line 132-133: "Solute and curvature effects are not considered for droplet growth by condensation". Does that mean that once the aerosols activates into a droplet, its growth is entirely proportional to the available supersaturation?

In most bulk and bin microphysics schemes (like $CL_{CCN}$ in this study), it is true that once aerosols are activated to cloud droplets, the growth of cloud droplets by condensation is calculated in a simple way $rdr/dt = Gs$, in which solute and curvature effects are not considered. However, to make a fair comparison with the haze-capable scheme ($CL_{Haze}$), we consider the solute and curvature effects for the growth of cloud droplets (radii larger than 1 μm) in both $CL_{CCN}$ and $CL_{Haze}$ schemes. The main difference between the $CL_{CCN}$ scheme and the $CL_{Haze}$ scheme is the way to handle droplet activation. In the $CL_{CCN}$ scheme, dry aerosols are activated to cloud droplets if s>$s_{crit}$ without resolving haze particles. In the $CL_{Haze}$ scheme, the growth of haze particles and the activation process is explicitly resolved. We have added more discussion in the manuscript:

"…*In this study, we consider the solute and curvature effects for the growth of cloud droplets (radii larger than 1 μm) in both $CL_{CCN}$ and $CL_{Haze}$ schemes. The main difference between the $CL_{CCN}$ scheme and the $CL_{Haze}$ scheme is the way to handle droplet activation as detailed above*.… "

2. Line 141: "evaporation can still occur due to turbulent supersaturation fluctuations". What is the source/origin of these fluctuations? How are they controlled? Do they play a role in broadening the droplet size spectra?

Turbulence in the chamber is forced by a warm bottom surface and cool top surface due to Rayleigh–Bénard convection. As long as we maintain the temperature difference between top and bottom, we can generate steady-state turbulence in the chamber. Turbulent mixing of air affected by different surfaces (e.g., bottom, top, side wall) leads to temperature and water vapor fluctuations, resulting in supersaturation fluctuation. It has been shown in our previous study (Chandrakar et al., 2016) that supersaturation can broaden droplet size distribution due to stochastic condensational growth in a turbulent environment.

3. Line 145: Is the removal by sedimentation uniquely due to gravity or also due to vertically oscillation motion within the chamber? Hence, does it affect all particles equally or it affects more the larger ones?

The motion of droplets in the chamber is affected by both advection and sedimentation. Advection transport of droplets in LES is size-independent, but there is no loss of droplets due to advection in our simulation (i.e., we set the flux of hydrometeors through any chamber surface to be zero). Particles (e.g., haze and cloud droplets) can only be removed out of the simulation though sedimentation and particle sedimentations are size dependent. Larger particles sediment faster due to their larger terminal velocities.

4. Does the sedimentation rate depend on the injection rate?

Yes, if the cloud reaches a steady state, the sedimentation rate should be equal to the aerosol injection rate.

5. Line 204: How is the turbulent mixing time defined in the context of the Pi-chamber?

We assume the transition between the slow and fast microphysics regimes occurs at Da~1, and thus we can estimate the mixing time scale based on the phase relaxation time at Da=1. The characteristic mixing time can also be estimated as $\tau_m = H/v_{air}$, where H is the chamber height, and $v_{air}$ is the characteristic air motion in the chamber. These two estimates in $\tau_m$ are not the same. We have added more discussion in the manuscript.

"…*The apparent transition between slow and fast regimes as shown in Fig. 1 provides an opportunity to estimate* $\tau_m$, *which is*  **about 70 s** *for our boundary conditions (e.g., 20 K difference in top and bottom temperature),* **if we assume the transition occurs at Da $\approx$ 1. However, this value is larger than the estimate via $\tau_m = H/v_{air}$. Here, H=1 m is the chamber height and $v_{air} \approx$ 0.1 ms$^{-1}$ is the characteristic air speed in the chamber based on LES, leading to $\tau_m$ on the order of 10 s. It is also larger than another estimate of $\tau_m = H^{2/3}/\varepsilon^{1/3} \approx$ 6 s, where $\varepsilon$ is the energy dissipation rate (about 0.005 m$^2$s$^{-3}$ from the simulation).**"

6. (5) represents particle removal, and it depends on the radius of the class of particles in question. Injection of aerosols also affects the number of particles, how is this accounted for in the full equation?

We guess the reviewer refers "the full equation" to the equation for the droplet size distribution. From the Eulerian point of view, as the reviewer said, droplet size distribution is affected by aerosol injection, condensational growth, and sedimentation. However, here, we simulate cloud microphysical processes in the box model from the Lagrangian point of view. Lagrangian particles are injected at a constant rate in a box. The size of the Lagrangian particle increases with time due to condensation, while the multiplicity of the Lagrangian particle decreases due to sedimentation. The initial multiplicity of each Lagrangian particle is the same, and it is related to the initial aerosol injection rate. Eq. 5 describes the loss of particles (i.e., multiplicity) due to sedimentation from the perspective of a Lagrangian particle, and it depends on the radius of that Lagrangian particle. In the end, the total droplet number concentration (i.e., the sum of multiplicity of all Lagrangian particles) in the box model can reach a steady state when the injection rate is balanced by the sedimentation rate.

7. The oscillations seem to be driven by aerosol injection rates and loss through sedimentation. If the system reaches a steady-state, I understand that injection and loss are compensated and, therefore, there is no net flux of particles. What, then, provokes oscillations, for example, in Figure 4 or 11?

This is a good question. It is true that the net droplet activation rate should be equal to the sedimentation rate in the steady state (e.g., slow and fast microphysics regimes). In the cloud oscillation regime, the net droplet activation rate averaged over one cycle should also be equal to the sedimentation rate, so that there is no net flux of particles. We add another section to discuss the physical origin of cloud oscillation (see our response to the major comment 2 above).

Reviewer 2:

This manuscript could be considered for publication after a major revision. The authors conducted a series of large-eddy simulations (LESs) of a cloud in a convection chamber using a haze-capable Eulerian-based bin microphysics scheme to explore haze-cloud interactions over a wide range of aerosol injection rates. They observed three microphysics regimes as they increased the aerosol injection rate: slow microphysics regime, fast microphysics regime, and cloud oscillation. The cloud oscillation is a new phenomenon being reported for the first time in this study. To understand the physical mechanism, they conducted a detailed analysis by introducing a box model. They also found that cloud collapse can occur if the side wall humidity is low. By solving a haze-only box model analytically, they also found the existence of a haze-only solution when aerosol injection rate is high.

As the LES using Twomey-type activation reproduced the slow and fast microphysics regimes but could not capture the cloud oscillation, they concluded that the haze-cloud interaction is critical in polluted conditions, but we could still use Twomey-type activation parameterizations for less polluted conditions.

The manuscript is relatively well written. However, the analysis is not comprehensive enough, and this makes the author's main conclusions not fully convincing. In particular, I see two major issues in this study:

1. They concluded that we could still use Twomey-type activation parameterizations for less polluted conditions. But, in my opinion, the conditions they tested are rather limited and the conclusion could be misleading.

2. I was confused that the author's interpretation regarding the cloud oscillation is sometimes not consistent with the plots they presented. A more careful analysis on supersaturation, supersaturation fluctuation, activation rate, deactivation rate, and sedimentation rate is desired. I believe it should foster a deeper understanding about the cloud oscillation phenomenon.

Please also see my more detailed comments provided below. I believe the quality of the study will be significantly enhanced if these points are addressed. I look forward to reading the revision of this manuscript.

We sincerely appreciate Prof. Shin-ichiro Shima for his time and efforts to provide detailed and insightful comments. We have addressed the reviewer's comments point-by-point below and made major revision of the manuscript based on those comments. Specifically, we conduct more simulations and data analysis to demonstrate the cloud oscillation phenomenon and explain the physical origin of cloud oscillation.

Major Comments

1) [request] P. 10 ll. 207-208 "The apparent transition between slow and fast regimes as shown in Fig. 1 provides an opportunity to estimate $\tau\_m$"

From the discussion on p. 16, it seems the authors are thinking that the transition point is $n\_in = 0.1$ $〚cm〛^(-3) s^(-1)$ and $\tau\_m \approx 70$ s, but it is indicated only implicitly on p. 10. Please clarify this point. Well, it should be possible to calculate the turbulent mixing time from the flow field directly, e.g., by calculating the integral time scale of the turbulence.

We now indicate the mixing time scale early in the manuscript based on the suggestion. The estimated mixing time scale based on Da=1 is larger than turbulent mixing time scale in the chamber based on LES. This might be because the simple theoretical model without turbulence cannot exactly represent the complex dynamic system. We have modified the manuscript accordingly.

"…*The apparent transition between slow and fast regimes as shown in Fig. 1 provides an opportunity to estimate* $\tau_m$*, which is*  **about 70 s** *for our boundary conditions (e.g., 20 K difference*

*in top and bottom temperature), **if we assume the transition occurs at Da $\approx$ 1. However, this value is larger than another estimate of $\tau_m = H/v_{air}$. Here, H=1 m is the chamber height and $v_{air} \approx 0.1$ ms$^{-1}$ is the characteristic air speed in the chamber based on LES, leading to $\tau_m$ on the order of 10 s. It is also larger than another estimate of $\tau_m = H^{2/3}/\varepsilon^{1/3} \approx 6$ s, where $\varepsilon$ is the energy dissipation rate (about 0.005 m$^2$s$^{-3}$ from the simulation).**"*

2) [request] Figure 4

Please also show the time evolution of total particle number ($N\_d + N\_(a/h)$), activation rate, deactivation rate, sedimentation rate, s, and $\sigma(s)$. This should provide more insight into understanding the mechanism.

We add the time evolution of total particle number, activation rate, deactivation rate, s, σ(s), and precipitation rate based on the suggestion (see Figures R1 and R2 below). The s and $\sigma(s)$ do provide more insight into understanding of cloud oscillation. We add more discussion in the manuscript.

"**…$N_T$ has a much smaller oscillation magnitude compared with $N_d$ and $N_h$, suggesting that the oscillation of $N_h$ and $N_d$ are out of phase.**…"

"***Figure 6 shows time series of domain-averaged activation rate ($R_{act}$), deactivation rate ($R_{deact}$), supersaturation (s), standard deviation of supersaturation ($\sigma_s$), and surface precipitation rate (P). Here surface precipitation refers to the sedimentation of cloud droplets at the bottom surface. Results show that oscillations of bulk cloud properties when using the $CL_{Haze}$ scheme shown in Fig. 5 are associated with oscillations of process rates, like $R_{act}$, $R_{deact}$, and P. It is interesting to see that s is close to $s_{crit}$ (about 0.08%) when using $CL_{Haze}$ scheme, while s decreases with $n_{in}$ and approaches 0 when using $CL_{CCN}$ scheme. This is because the cloud system is buffered by huge amount of cloud droplets in the polluted condition and s should be close to the equilibrium supersaturation over droplets. Such equilibrium supersaturation is $s_{crit}$ when using the $CL_{Haze}$ scheme where solute and curvature effects are considered, but it is 0 when using the $CL_{CCN}$ scheme. Because $\sigma_s$ is much smaller than $s_{crit}$ at high injection rates, droplet activation is mainly controlled by the mean s. The oscillation of s around the $s_{crit}$ leads to the oscillation of droplet activation, and further causes the oscillation of cloud properties.***"

[Figure]

**Figure R1 (Figure 5 in the revision):** Time series of domain-averaged $q_l$ (first row), $N_d$ (second row), $N_a$ or $N_h$ (third row), **$N_T$ (fourth row),** and $r_d$ ( **fifth** row) for five different $n_{in}$: 10, 20, 30, 40, and 50 cm$^{-3}$s$^{-1}$. The  **light blue** line in the first row represents the cloud water mixing ratio ($q_c$) when using the CL$_{Haze}$ scheme.

[Figure]

**Figure R2 (Figure 6 in the revision): Same cases in Fig. 5, time series of domain-averaged $R_{act}$ (first row), $R_{deact}$ (second row), s (third row), $\sigma_s$ (fourth row), and surface precipitation rate P (fifth row) for five different $n_{in}$: 10, 20, 30, 40, and 50 cm$^{-3}$s$^{-1}$.**

3) [request] P. 11 l. 220 "Note that Na increases with time for $n\_in \geq 10$ cm$^{-3}$s$^{-1}$."

Do you mean $n\_in \geq 40$ cm$^{-3}$s$^{-1}$? Please clarify.

Yes, we mean $n\_in \geq 40$ cm$^{-3}$s$^{-1}$. We have corrected it in the revision.

"*Na increases with time for $n_{in} \geq$  40 cm$^{-3}$s$^{-1}$...*"

4) [suggestion] P. 11 l. 224 "The oscillation period increases as nin increases, ..."

It should be also informative to point out that the oscillation amplitude increases as nin increases.

Thanks for the suggestion. We have modified the manuscript accordingly.

"***Meanwhile, the oscillation amplitude increases with $n_{in}$.***"

5) [request] Figure 5i

For a better comparison, please use the same color bar as in Figure 6.

We have modified the figure based on the suggestion. Figure 5i now looks like this:

[Figure]

The color bar is now the same as used in Figure 6.

6) [question] P. 11 ll. 234–235 "The sharp increase in Nd (Fig. 5b) corresponds to a larger activation rate (Fig. 5c) due to the enhanced supersaturation (Fig. 5i), …"

From Figs. 5c and 5f, it looks like the mean $R_{deact}$ is always larger than mean $R_{act}$, though on average $R_{act} - R_{deact} = n_{in}$ has to be satisfied. Why is it?

To better show the low value of $R_{act}$ and $R_{deact}$, we set $R_{act}$ and $R_{deact}$ to 240 cm$^{-3}$s$^{-1}$ if their values are larger than it when plotting Fig. 5c and Fig. 5f. as shown below.

[Figure]

Based on the reviewer's suggestion, we add a new figure to show the time series of domain-averaged $R_{act}$ and $R_{deact}$ (see the first and second rows in Figure R2 above). The peak of $R_{act}$ in one cycle is over $10^3$ cm$^{-3}$s$^{-1}$ while the peak of $R_{deact}$ is smaller. So, although the net activation rate ($R_{act}$ - $R_{deact}$) at one time can be negative, the net activation rate averaged over one cycle is still positive. We add more discussion in the manuscript:

*"…To better show the low value of $R_{act}$ and $R_{deact}$, we constrain the range of $R_{act}$ and $R_{deact}$ to values below 240 cm$^{-3}$s$^{-1}$ if their values are larger than it when plotting Fig. 7 c and f. It can be seen that deactivation occurs in a much larger region (i.e., outside the top and bottom surfaces) and over a longer time period within one cycle. However, the peak of $R_{act}$ is larger than the peak of $R_{deact}$ (see Fig. 6 first and second rows). The net activation rate ($R_{act}$-$R_{deact}$) averaged over one cycle should still be positive, so that sedimentation is balanced by the net activation."*

7) [question] P. 11 l. 235 "…, while the decrease in Nd corresponds to a larger deactivation rate and a smaller supersaturation."

If we compare Figs. 5f and 5i, the deactivation rate is larger even when supersaturation is larger, which is counterintuitive. Is this due to supersaturation fluctuation?

On a related note, is removal by sedimentation much smaller than $R_{deact}$ and $R_{act}$?

It is true that it looks like that the deactivation rate is larger when supersaturation is larger from the figure. However, s is still smaller than $s_{crit}$. We change the unit of s in Fig. 5i from % to ‰ (see the figure below) to better illustrate this point. Although there is a large region of high s band (e.g., at around 9.3 hour), its value is still less than $s_{crit}$ (8 ‰) except close to the top and bottom surfaces. The deactivation in the bulk region is mainly driven by the mean s (when s<$s_{crit}$, due to curvature effect, see Fig.1 in Nenes et al., 2001). Deactivation is not driven by supersaturation fluctuation, because as shown in Figure R2 (the fourth row), the supersaturation fluctuation is much smaller than $s_{crit}$ in the polluted condition due to the buffering effect of cloud droplets. Yes, the removal rate is much smaller than $R_{act}$ and $R_{deact}$. In the steady state, the injection rate equals to the net activation rate ($R_{act}$-$R_{deact}$), and equals to the sedimentation rate, averaged over one cycle.


From the observation described in this section, it is not clear why cloud oscillation does not occur when using the CL_CCN scheme.

Time series of s and $\sigma_s$ shown in Figure R2 provide more insights of why cloud oscillation does not occur when using the $CL_{CCN}$ scheme. The direct reason for cloud oscillation is that s oscillates around $s_{crit}$ when using the $CL_{Haze}$ scheme. To be clear, cloud oscillation mentioned in this study represents the oscillation of cloud bulk statistical properties. It is the oscillation of the whole well-mixed cloud system, not an individual droplet. The physical origin of cloud oscillation is due to the non-linear interactions between haze and cloud droplets in a dynamic system: (1) First, there is a forcing in the system to maintain the supersaturation; (2) When $s>s_{crit}$, huge number of droplets are activated and the consumption of water vapor due to droplet condensational growth leads to $s<s_{crit}$; (3) Under $s<s_{crit}$ condition, droplet activation is suppressed and droplet concentration decreases due to droplet deactivation and sedimentation; (4) Meanwhile, haze number concentration increases due to continuously aerosol injection and droplet deactivation due to supersaturation fluctuation. (5) s increases as the decrease of the sink of water vapor due to fewer cloud droplets and more haze particles, and when $s>s_{crit}$, another cycle starts.

In contrast, s approaches 0 when using the $CL_{CCN}$ scheme. In addition, both s and $\sigma_s$ are smaller than $s_{crit}$, suggesting that droplet activation is strongly suppressed in the bulk region. Droplet activation occurs close to the top and bottom surfaces where s in the local region is larger than $s_{crit}$, while droplet deactivation occurs in the bulk region, same as when using the $CL_{Haze}$ scheme (See Fig. R2). However, the main difference is that if droplets are deactivated, they go back to the dry aerosol category when using the $CL_{CCN}$ scheme, while in contrast, when using the $CL_{Haze}$ scheme, droplets stay as haze particles which can still consume water vapor and contribute to liquid water content.

We added a new subsection "3.2.3 Origin of cloud oscillation" to discuss about it.

*"**3.2.3 Origin of cloud oscillation***

*Results from LES and box models show the existence of cloud oscillation at high $n_{in}$, indicating that cloud oscillation is physically plausible, not due to numerical artifact. In this subsection, we discuss the physical origin of cloud oscillation and explain why the $CL_{Haze}$ scheme can simulate cloud oscillation, while the $CL_{CCN}$ scheme fails.*

*Time series of s shown in Figs. 6 and 11 provide more physical insights of cloud oscillation. The direct reason for cloud oscillation is that s oscillates around $s_{crit}$ when using the $CL_{Haze}$ scheme. To be clear, cloud oscillation mentioned in this study represents the oscillation of cloud bulk statistical properties. It is the oscillation of the whole well-mixed cloud system, not an individual droplet. The physical origin of cloud oscillation is due to the non-linear interactions between populations of haze and cloud droplets in a dynamic system: (1) First, the supersaturation s in the system is very close to $s_{crit}$, and most of the time $s<s_{crit}$. This can happen in a heavily polluted condition where there are many haze particles. (2) There is a forcing in the system to maintain the supersaturation. In the Pi chamber, the forcing is due to the temperature difference between top and bottom surfaces. In the real atmosphere, the forcing can be due to adiabatic cooling (e.g., in a rising cloud parcel) or radiative cooling (e.g., radiation fog). (3) When $s>s_{crit}$, huge number of haze particles activate to cloud droplets and the consumption of water vapor due to droplet condensational growth leads to $s<s_{crit}$; (4) Under $s<s_{crit}$ condition, droplet activation is suppressed and droplet concentration decreases due to droplet deactivation and sedimentation; (5) Meanwhile, haze number concentration increasse due to continuously aerosol injection and droplet deactivation. (6) s increases with the decrease of the sink of water vapor due to fewer cloud droplets and more haze particles, and when $s>s_{crit}$, another cycle starts. In contrast, s approaches 0 when using the $CL_{CCN}$ scheme (black line in the third row of Fig. 6), suggesting that droplet activation is strongly suppressed in the bulk region.*

*Additionally, $\sigma_s$ decreases with $n_{in}$ and approaches 0 due to the buffering effect of cloud droplets under polluted conditions (Fig.6 fourth row). This suggests that droplet activation is controlled by the mean supersaturation instead of supersaturation fluctuation. This is why even though turbulence is not considered, the box model (Fig. 9) and the theoretical model (developed in Shaw et al., 2023) can still predict the scaling relationships in fast and slow microphysics regimes that are consistent with large-eddy simulations (Figure 1) and Pi chamber experiments (Figure 7 in Shaw et al., 2023). The nice performance of the theoretical model and the box model suggests that turbulence is not the direct factor in generating various microphysics regimes, including provoking cloud oscillation. As long as the cloud is well mixed (due to turbulence), various microphysics regimes (e.g., slow, fast, oscillation) can occur under different aerosol injection rates (for monodisperse aerosols like in this study).*

*It is interesting to see that droplet deactivation can still occur even though $s$ is always positive in the box model (Fig. 11). This is also likely to be true in LES when using $CL_{Haze}$ scheme, in which s oscillates around $s_{crit}$ and $\sigma_s \ll s_{crit}$. One question is what drives droplet deactivation in a supersaturated environment. Here, droplet deactivation occurs due to the curvature effect. Although haze particles can be activated to droplets when $s > s_{crit}$, the subsequent decrease of s (like s oscillation in our case) can lead to droplet deactivation when s is smaller than the saturated saturation ratio over small cloud droplets (see green line in Fig. 1 of Nenes et al., 2001). In addition, there is one difference in handling droplet deactivation between the $CL_{CCN}$ and the $CL_{Haze}$ schemes. If droplets are deactivated, they go back to the dry aerosol category when using the $CL_{CCN}$ scheme. When using the $CL_{Haze}$ scheme, however, droplets stay as haze particles which can still consume water vapor and contribute to liquid water content. The latter has feedback in s which is critical to trigger cloud oscillation that we will discuss next.*

*When $s < s_{crit}$, droplet activation is suppressed in the bulk region. This is true for both $CL_{Haze}$ and $CL_{CCN}$ schemes. However, when using the $CL_{Haze}$ scheme, the contribution of haze water content to the total liquid water content increases under this condition ($s < s_{crit}$) due to continuous aerosol injection and droplet deactivation (as discussed above). The sink of water vapor via condensational growth decreases due to the decrease of cloud droplet concentration, which can lead to an increase in s, considering that the source of water vapor from chamber surfaces is constant. When $s > s_{crit}$, droplet activation is active again. In contrast, when using the $CL_{CCN}$ scheme, haze water content is not considered, and cloud droplet content is equivalent to liquid water content. In addition, both s and $\sigma_s$ are buffered to approach 0 under polluted conditions, and there is no restoring force to increase s.*"

12) [suggestion] Eqs. 1, 2, and 4

The $\llbracket dq \rrbracket\_l/dt$ here represents the change of $q_l$ through diffusional growth, but droplet removal by sedimentation also changes $q_l$. To avoid confusion, the authors should clarify this point, e.g., by using a subscript: $\llbracket dq \rrbracket\_l/dt|\_{diff}$.

We have modified Eqs. 1, 2, and 4 based on the reviewer's suggestion.

13) [question] Eq. 3

If I understand correctly, the supersaturation fluctuation is not considered in this box model, right? Is it not important for the phenomenon?

Yes, supersaturation fluctuation is not considered in the box model. We add another subsection to discuss more details of the physical origin of cloud oscillation. (see our response to point 11 in more details).

14) [question] P. 16 l. 299 "T0 and qv0 are set to be 290 K and 13.9 g kg−1, …"

What is the corresponding supersaturation $s_0$?

The corresponding s0 is 15%. We modify the text accordingly.

"…, ***corresponding to a supersaturation of 15% in the absence of all hydrometeors,*** …"

15) [request] P.16 l.302 "... the estimated $\tau_m$ for Da = 1 based on LES results …"

Please clarify that the estimated $\tau_m$ was 70 s.

We have clarified the value in that sentence.

"*Note that the value of $\tau_m$ used here is not exactly the same as the estimated $\tau_m$ **(~70 s)** for Da=1 based on LES results above…*"

16) [request] Figure 8

Please also show the time evolution of total particle number ($N_d+N_h$), activation rate, deactivation rate, sedimentation rate, and s.

We now add the time evolution of total particle number concentration, activation rate, deactivation rate, s, $\sigma_s$, and sedimentation rate as requested.

[Figure]

**Figure R3 (Figure 10 in the revision):** Time series of $q_l$ (first row), $N_d$ (second row), $N_h$ (third row), **$N_t$ (fourth row)** and $r_d$ (fifth row) from a box model using a Lagrangian microphysics approach for the five largest $n_{in}$: 10, 20, 30, 40, 50 cm$^{-3}$s$^{-1}$.

[Figure]

**Figure R4 (Figure 11 in the revision): Similar to Fig. 10, time series of s (first row), $R_{act}$ (second row), $R_{deact}$ (third row), and $R_{sed}$ (fourth row) from a box model using a Lagrangian microphysics approach for the five largest $n_{in}$: 10, 20, 30, 40, 50 cm$^{-3}$s$^{-1}$.**

17) [question] P.16 l.316 "In contrast, simulations using the CL_CCN scheme do not show oscillations …"

What will happen if we use Twomey activation for this box model? Fig. 4 revealed that LES with CL_CCN does not show oscillation, but it seems to me nothing prevents the Twomey box model from exhibiting oscillation?

The main purse of conducing the box model in our study is to confirm that the cloud oscillation from LES is not due to numerical artifacts (e.g., numerical diffusion when using the bin microphysics scheme). The box model we use here employs the Lagrangian particle approach and the ODE solver, which can simulate the growth of haze particles and cloud droplets very accurately (i.e., with the least parameterization of cloud microphysical processes, such as droplet activation and condensational growth). Adding Twomey activation in the current box model is not straightforward and it will introduce additional assumptions (e.g., how to handle the growth of haze particles, do we consider the solute and curvature effects…). Based on the reviewer's other comments, we have conducted more data analysis, added new figures and a new section (3.2.3) to illustrate the origin of cloud oscillation. We hope we make it clear about why the LES with CL$_{CCN}$ does not show oscillation now.

18) [question] Sec. 3.2.2 "Cloud oscillation in a box model"

Additional questions about this section:

● Because supersaturation fluctuation is not considered in this model, always holds, and deactivation of droplets does not occur in this model. Is this correct?

Supersaturation fluctuation is not considered in the box model. But droplet deactivation can still occur due to curvature effect (see Fig.1 in Nenes et al., 2001). We add more discussion in the manuscript.

*"It is interesting to see that droplet deactivation can still occur even though s is always positive in the box model (Fig. 11). One question is what drives droplet deactivation in a supersaturated environment. Here, droplet deactivation occurs due to the curvature effect (Nenes et al., 2001). However, there is one difference in handling droplet deactivation between the $CL_{CCN}$ and the $CL_{Haze}$ scheme. If droplets are deactivated, they go back to the dry aerosol category when using the $CL_{CCN}$ scheme. When using the $CL_{Haze}$ scheme, however, droplets stay as haze particles which can still consume water vapor and contribute to liquid water content. The latter has feedback in s which is critical to trigger cloud oscillation that we will discuss next."*

● Instead, the droplets are removed from the system only by sedimentation in the box model. Is this correct?

Yes, droplets are removed only by sedimentation in the box model.

● The oscillation amplitude of $N_d$ in Fig. 8 is smaller than that in Fig. 4. Is this because of the absence of deactivation in the box model?

No, deactivation also occurs in the box model due to the curvature effect (see the third row in Fig. R4).

● Because supersaturation fluctuation is not considered in this model, almost all haze particles should be activated when $s>s_{crit}$. This is the reason why $N_h$ decreases to almost 0 in Fig. 8 (though this is not happening in Fig. 4). Is this correct?

Yes, this is correct.

19) [question] P.18 l.344 "..., the oscillation frequency approaches zero)?

Why do you think the frequency approaches zero when switching to the haze-only regime?

In dynamical systems theory, there are various types of bifurcations that are responsible for the onset of oscillation. If the frequency approaches zero, it suggests it is an "infinite period bifurcation". On the other hand, if the oscillatory solution arises when a fixed point (haze-only solution) is destabilized, it is a Hopf bifurcation. Then, the frequency is finite at the bifurcation point. See, e.g., Strogatz (2014).

We agree with the reviewer's comments and remove our hypothesis there. We also add some discussion in the end of that subsection.

*"…Would there be another regime in which there are only haze particles and no cloud droplets ?…"*

*"…Of cause, this is only our conjecture, not a formal proof. Further efforts are needed to understand the onset of oscillation, the transition between oscillation regime and haze-only regime, and the stability of the haze-only regime."*

20) [suggestion] Eqs. 7–11

Again, it should be clarified that only the contribution via diffusional growth is considered, e.g., by using a subscript: $⟦dq⟧\_l/dt|\_diff$.

In particular, Eq. 11 is confusing if $⟦dq⟧\_l/dt$ is used on the l.h.s.; if we take it literally, it indicates grows exponentially! But, of course, the correct meaning is $⟦dq⟧\_l/dt|\_diff = q\_l/\tau\_sed$. I would suggest the expression $n\_in = N\_h/\tau\_sed$.as an equivalent but more intuitive alternative.

We modify Eqs. 7-11 based on the suggestion and add a clarification in the manuscript.

*"$dq_l/dt|_{diff}$ in Eqs. 7 and 8 indicates that only the contribution via diffusional growth is considered here…."*

21) [question] Eq. 10

I think the use of this formula is not appropriate for this analysis, because $r\_eq \to \infty$ as RH->1. How about simply assuming $r\_eq$ = const.?

We have updated the equation such that it also works for RH=100% based on Eq. 10 of Lewis (2019).

*"Here $r_{eq}$ is the equilibrium haze particle radius at a given s $<$ $s_{crit}$,* **which depends on the environmental fractional relative humidity (RH $\equiv$ 1+s) and on properties of the substance.**  **We assume that particles reach their equilibrium size within a very short time. $r_{eq}$ can be expressed as a function of RH for values near but smaller than unity based on Eq. 10 of Lewis (2019), where the constants are those for sodium chloride**:

$$\frac{r_{eq}}{r_{dry}} = \frac{1.04}{\left[1 - h + \left(\frac{0.99 \text{ nm}}{r_{dry}}\right)^{3/2}\right]^{1/3}}.$$

**This expression is accurate to within 5 % for values of h between 99% and 100%, and for $r_{dry} > 10$ nm.**"

22) [question] Figure 10

How are the left ends of the two lines determined? Are they corresponding to RH=1? If so, Eq. 10 should not be used because it is not accurate when approaches 1.

Yes, the left ends of the two lines are determined at RH=100%. Equation 10 has been updated based on Lewis (2019). See details in our response above. We also add clarification of this point in the manuscript.

"…**The left ends of the two lines in Fig. 10 are determined at RH=100%**…"

23) [request] P.21 l.374 "…, so that s $<$ $s_{crit}$ all the time."

Because of the use of Eq. 10, the analysis is valid only for s<0 ($<s_{crit}$). Please clarify this point.

Yes, it should be s<0. We have corrected the statement.

"…, *so that*  **s<0** *all the time.*"

24) [comment] Sec. 3.2.4 "Haze-only regime"

The existence of the haze-only solution is presented in this section, but it does not guarantee it is stable.

We appreciate the reviewer for bringing up this good point. We agree with the reviewer that we do not show whether the haze-only regime is stable or not analytically. But we expect the steady state in the haze-only regime is stable for a given $n_{in}$. This is because in the steady state, the aerosol injection rate should be equal to the sedimentation rate of haze particles,

$$n_{in} = \frac{N_h}{\tau_{sed}}$$

If there is a positive (or negative) perturbation of $N_h$, the sedimentation rate will increase (or decrease), leading to a net decreasing (or increasing) tendency of $N_h$ for a given $n_{in}$. This negative feedback is trying to bring $N_h$ back to its steady state value. We add more discussion in the manuscript to address this point.

*"So far, we have demonstrated the existence of the haze-only microphysics regime in an idealized scenario. One question is whether the haze-only regime is stable. We expect that the steady state in the haze-only regime is stable for a given $n_{in}$. This is because aerosol injection rate should be equal to the sedimentation rate of haze particles in the steady state (see Eq. 11). If there is a positive (or negative) perturbation of $N_h$, the sedimentation rate would increase (or decrease), leading to a net decreasing (or increasing) tendency in $N_h$ for a given $n_{in}$. This feedback is trying to bring $N_h$ back to its steady state value. Of course, this is only our conjecture, not a formal proof. Further efforts are needed to understand the onset of oscillation, the transition between oscillation regime and haze-only regime, and the stability of the haze-only regime."*

25) [request] P.23 l.433 "deactivation (s < scrit)"

Do you mean s<0?

In both CL_haze and CL_CCN, deactivation occurs only when the supersaturation is locally smaller than zero. Hence, if there is no fluctuation, $0<s<s_{crit}$ (s represents mean supersaturation) does not induce any deactivation. (More precisely speaking, if all the activated droplets are large enough.) If the supersaturation fluctuation is large, deactivation can occur locally even when $s_{cirt}<s$.

Based on the new data analysis, we show that $\sigma_s<<s_{crit}$ in the cloud oscillation regime, while s oscillates around $s_{crit}$, which is the main reason to generate cloud oscillation. We have modified the text accordingly.

*"… Specifically, in the cloud oscillation regime, s oscillates around $s_{crit}$ and $\sigma_s << s_{crit}$, because the cloud system is buffered by huge amount of haze particles and cloud droplets. Droplet activation is controlled by the mean supersaturation rather than supersaturation fluctuation, while droplet deactivation can still occur in supersaturated environment ($0<s<s_{crit}$) due to curvature effect. The oscillation of s around $s_{crit}$ leads to the oscillation of droplet activation and deactivation, and further causes the oscillation of cloud properties. …leading to synchronized activation ($s>s_{crit}$) …"*

26) [request] P.24 ll.443–444 "Haze-cloud oscillation is more likely to occur under conditions of weak supersaturation forcing, …"

Why do you think so? Please elaborate.

We modify the sentences in the following way to make it clear.

*" Our results suggest that haze-cloud interactions are very important when air supersaturation is close to the critical supersaturation of aerosols.  This condition happens in the Pi chamber at high aerosol injection rates as shown in this study, and it  can also occur in the atmosphere, for example, when cloud or fog is close to the source of intense natural and anthropogenic aerosol emissions…."*

27) [comment] P.24 l.455 "Our results suggest that haze-cloud interactions are very important especially in polluted conditions."

The authors suggest that we could still use Twomey-type activation parameterizations for less polluted conditions, but I am not fully convinced. NaCl aerosol particles with a dry radius of 62.5 nm are considered in this study, but they are relatively small. I think the haze-cloud interaction should be more important for larger aerosol particles because the equilibrium wet radius gets larger and the activation/deactivation time scale gets longer. In addition, aerosols considered are monodisperse in this study, but I also think that haze-cloud interaction is more important for polydisperse aerosols. Please see, e.g., Fig.5 of Richter et al. (2021).

These limitations of this study should be discussed and emphasized more.

We modify our manuscript accordingly based on the reviewer's constructive comments. The limitations of this study are discussed and emphasized in that paragraph.

"* Our results suggest that haze-cloud interactions are very important when air supersaturation is close to the critical supersaturation of aerosols.  Such condition is possible in the Pi chamber at high aerosol injection rates as shown in this study, and it  can also occur in the atmosphere, for example, when cloud or fog is close to the source of intense natural and anthropogenic aerosol emissions… Also note that monodisperse aerosol with a dry radius of 62.5 nm is used in this study. We expect haze-cloud interaction might be more important for larger aerosol particles because their critical supersaturation gets smaller, their equilibrium wet radius gets larger, and the activation/deactivation time scale could get longer (Hoffmann, 2016). In addition, aerosol particles in nature vary in size and compositions, and haze-cloud interactions might be more important for polydisperse aerosols (see Fig. 5 in Richter et al., 2021), which is also worth exploring in the future*…."

Minor Comments

28) [request] P.2 ll.54–55 "Shaw et al. (2023)" Because polydisperse aerosol injection is discussed in the previous sentence, please clarify that monodisperse aerosol injection was assumed in Shaw et al. (2023).

We clarify this point in the manuscript.

"…***Recently,*** *Shaw et al. (2023) developed a theoretical model to describe the microphysical state of **well mixed monodisperse droplets** in cloudy Rayleigh-Benard convection*…."

29) [request] P.7 Fig. 1

Please also show the standard deviation of supersaturation.

What is the definition of droplet mean radius and droplet number concentration for the CL_Haze scheme? Are the haze droplets included in the population? Please clarify.

Figure below (current Figure 2 in the revision) shows supersaturation and the standard deviation of supersaturation at different $n_{in}$ ($\leq 5$ cm$^{-3}$s$^{-1}$). Results show that $\sigma_s$ is independent of $n_{in}$ in the slow microphysics regime, and it first increases then decreases as the increase of $n_{in}$ in the fast microphysics regime. The general decreasing trend of s and $\sigma_s$ with $n_{in}$ is mainly due to the buffering effect of cloud droplets: polluted clouds can response faster to the change of environment (compared with clean clouds) and thus has less fluctuations.

The new figure provides a connection between microphysics regimes and activation regimes. The slow microphysics regime is in the mean-supersaturation-dominated activation regime where s >> s$_{cirt}$. The fast microphysics regime covers the supersaturation-fluctuation-influenced and supersaturation-dominated activation regimes where s ~ s$_{cirt}$. We add more discussion in the manuscript.

"*Two microphysics regimes (slow and fast) are observed in Figure 1, and the impact of $n_{in}$ on the mean supersaturation s and its standard deviation $\sigma_S$ (see Figure 3) indicates the physical origin of the two microphysics regimes, and its connection to various activation regimes….Based on the definition in Prabhakaran et al. (2020), the cloud is in the mean-supersaturation-dominated activation regime where $s >> s_{crit}$.*"

"…*Based on the definition in Prabhakaran et al. (2020), The cloud is in the supersaturation-fluctuation-influenced activation regime ($s>s_{crit}$ and $\sigma_s>s_{crit}$) or supersaturation-fluctuation-dominated activation regime ($s<s_{crit}$ and $\sigma_s>s_{crit}$), but the latter is barely observed in our results .*"

"*The scaling laws for $N_d$ and $q_l$ do not work well for $n_{in}>10.0$ cm$^{-3}$s$^{-1}$ when using the $CL_{Haze}$ scheme (Figure 1 b and c). Also note that both s and $\sigma_s$ are smaller than $s_{crit}$ at these high aerosol injection rates, suggesting that droplet activation is strongly suppressed. It is interesting to see that s approaches a value which is slightly smaller than $s_{crit}$ when using the $CL_{Haze}$ scheme, while in contrast, s continuously decreases with $n_{in}$ and approaches 0 when using the $CL_{CCN}$ scheme. This is because the cloud system is buffered by huge amount of cloud droplets in the polluted condition and s should be close to the equilibrium supersaturation over droplets (which is $s_{crit}$ when using the $CL_{Haze}$ scheme where solute and curvature effects are considered, or 0 when using the $CL_{CCN}$ scheme). This regime turns out to be very important for haze-cloud interactions that will be explored in the following section.*"

Haze particles are not included when calculating the cloud properties (e.g., droplet mean radius and droplet number concentration. We have stated in the main text, and to make it clear, we add clarification in Figure 1 caption.

"…*Note that we only consider cloud droplets whose radii larger than 1 µm to calculate $r_d$ and $N_d$ here.*"

[Figure]

**Figure R5: Similar to Figure 1 in the main text, spatial- (over the whole domain) and temporal-averaged (in the second half an hour) supersaturation (s) and standard deviation of supersaturation ($\sigma_s$) at various aerosol injection rates ($n_{in}$). The horizontal dashed line indicates the critical supersaturation of injected aerosols (0.08%).**

30) [comment] P.9 Table 2 and elsewhere "Na/Nh"

This looks like "Na divided by Nh" and I believe it is confusing to the readers. How about simply writing "Na, Nh" or "Na or Nh"?

We change "*Na/Nh*" to "**Na or Nh**".

31) [comment] P.9 Table 2

Please consider including units in the headers to enhance readability.

We modify Table 2 based on the suggestion.

32) [question] P.10 ll.198–199 "Note that the net activation rate (Ract−Rdeact) is close to nin for each case suggesting that the cloud reaches a quasi-steady state."

In a quasi-steady state, the removal rate by sedimentation should be equal to the net activation rate and injection rate. Did you confirm this? To put it another way, how did you confirm that one hour is sufficient to reach a quasi-steady state?

The last two columns in Table 2 list $R_{act}$ and $R_{deact}$ in for cases when $n_{in} \leq 5$ cm$^{-3}$s$^{-1}$. The net activation rate ($R_{act}$ - $R_{deact}$) is very close to $n_{in}$ (the first column in Table 2). Meanwhile, Figure 4 (first column) shows that cloud can reach a quasi-steady state within one hour based on 10-hours simulation for $n_{in}$=10 cm$^{-3}$s$^{-1}$. Based on our previous chamber simulation results (e.g., Yang et al., 2022; 2023), the cloud can reaches a quasi-steady state faster for lower aerosol injection rates, so we know one hour is sufficient to reach a quasi-steady state for $n_{in} \leq 5$ cm$^{-3}$s$^{-1}$. We have added more text in the manuscript to make it clear.

"…*Note that the net activation rate ($R_{act}$-$R_{deact}$, **the last two columns in Table 2**) is close to $n_{in}$ (**the first column in Table 2**) for each case suggesting that the cloud reaches a quasi-steady state.…*"

33) [comment] P.15 l.269 "Each particle represents numerous real particles per unit volume. We refer to this as multiplicity, …"

Note that multiplicity is defined differently in Shima et al. (2009).

We add clarification of this point in the manuscript.

"…***Note that the multiplicity in this study is different from that in Lagrangian microphysics scheme (e.g., Shima et al., 2009; Hoffmann et al., 2015) in which it represents multiple number (instead of concentration) of identical droplets represented by the Lagrangian particle/superdroplet***…"

34) [request] P.23 l.426 "But they do not capture the distribution properties …"

Please clarify that "they" represents "analytical estimates".

We change "*they*" to "***those analytical estimates***" based on the suggestion.

Typo

35) P.3 l.59 response -> respond

Corrected.

36) Eq. 4

G is not needed?

Yes, G is not needed. The equation is corrected.

37) Eq. 5

If $\delta n_i$ is representing the decreased amount of multiplicity, Eq. 5 has to be

$$\delta n_i = n_i \left( 1 - \exp \left[ -\frac{\delta t}{\tau_{sed}(r_i)} \right] \right) \approx n_i \frac{\delta t}{\tau_{sed}(r_i)}.$$

Corrected.

38) P.21 l.385 "following by Thomas et al." -> "following Thomas et al."

Corrected.

39) P.22 Eq. 12

This must be $\delta N\_h = N\_h\, \Delta t / t\_wl$.

Corrected.

---

## Author Response (AR3)

We appreciate the reviewers' and the editor's time and effort spent on our revised manuscript. We are glad to see that both reviewers are highly supportive of the revision. For further minor comments from the reviewer and the editor, we have addressed them carefully below.

From the reviewer 2:

All the points I raised in my previous review have been thoroughly addressed in the revised manuscript. The mechanism behind the cloud oscillation phenomenon is now much clearer. I believe the manuscript is almost ready for publication, with one exception. The authors mention that deactivation can still occur even when s > 0. While this is correct, it only occurs conditionally. As the figure below indicates, the size of the droplet has to be smaller than the unstable fixed point denoted by the white circle. If I understand correctly, this occurs when the phase relaxation time is shorter than the droplet activation time (which I think is also closely relevant to the onset of the catastrophe discussed in Arabas and Shima (2017)). I suggest that the authors expand their discussion on this point a bit. A further review of the revised manuscript will not be necessary.

We totally agree with the reviewer's comments about why droplets deactivate when s>0. It is consistent with ours, but it is interpreted in a different way. We extend our discussion based on the reviewer's comments in the revised manuscript.

"...***Droplet deactivation in supersaturated conditions occurs when the phase relaxation time is much shorter than the droplet activation time. This phenomenon is closely relevant to the onset of the catastrophe discussed in Arabas and Shima (2017)***...."

From the editor:

Thank you for the revisions of this fascinating manuscript using the Pi chamber. The reviewers are now both highly supportive with only a couple recommendations I hope you will consider.

I confess to one concern about the work. The results are highly interesting, of course, especially as an important reference. However an immediate question that springs to mind is about their applicability to natural clouds where the boundary conditions are changing over timescales of perhaps 10 minutes. Would the regimes identified still be as clear?

We appreciate the editor for raising this good point. We add some discussions about it in the manuscript.

"...***Furthermore, unlike well-controlled environmental conditions in the cloud chamber, the boundary conditions of natural clouds or fogs change over time, which would also affect the microphysical regimes. The impact of haze-cloud interactions under real cloud conditions is worth exploring in the future***."